# NEUROBASKET: INTERPRETING NEURON RESPONSES WITH SEMANTIC BASKETS

## ABSTRACT

Deep neural networks excel across domains, yet their internal representations remain opaque. Prior approaches based on single neurons or non-hierarchical groups are limited by the distributed nature of concept encoding. We introduce Neurobasket, a framework that constructs semantically coherent multi-neuron groups through hierarchical clustering and natural language grounding. Neurobaskets enable set-operation–based analysis (union/difference), with unions revealing shared abstractions and differences highlighting discriminative cues. Experiments across convolutional and transformer models, trained on diverse datasets, show that neurobaskets yield stable and semantically aligned sets, while capturing prediction-relevant pathways. Qualitative visualizations further showed that grouped neurons correspond to coherent and localized concepts. Overall, Neurobasket provides a structured and compositional view of neural representations, extending beyond unit-centric or non-hierarchical explanations.

## 1 INTRODUCTION

Deep neural networks (DNNs) have achieved remarkable success in computer vision, natural language processing, and multimodal reasoning. Yet their internal mechanisms remain opaque, raising concerns about reliability and trustworthiness. This has fueled growing interest in explainable AI (XAI), where the goal is to understand how neural networks represent and process information.

A key line of research in XAI considers the neuron as the basic unit of explanation. Early studies such as *NetDissect* (Bau et al., 2017) and its follow-ups (Zhou et al., 2018; 2021; Oikarinen & Weng, 2023; Ahn et al., 2024) showed that individual neurons can sometimes align with human-interpretable concepts, initiating the study of Single-Neuron Explanations (SNE). However, SNE often struggles in modern architectures where representations are highly distributed: only a fraction of neurons align cleanly, many encode compositional mixtures of concepts (Mu & Andreas, 2020), and superposition of multiple features within the same unit is common (Elhage et al., 2022). As a result, unit-level explanations may be fragmented or only weakly connected to model predictions.

This motivates Multi-Neuron Explanations (MNE), which aim to capture coordinated activity across groups of neurons. Recent approaches such as FALCON (Kalibhat et al., 2023) and NeurFlow (Cao et al., 2025) demonstrate the potential of analyzing neuron groups. At the same time, they highlight a fundamental difficulty: determining which neurons should be grouped is non-trivial, especially across layers and levels of abstraction. Existing approaches typically rely on non-hierarchical groupings or one-to-one mappings between groups and concepts, leaving open the question of how to organize multi-neuron sets in a structured and semantically coherent way.

Figure 1 illustrates this central challenge. Even if individual neurons can be assigned meaningful concepts, without understanding the core set of neurons for coherent explanations, it merely produces redundancy (e.g., greenish yellow and green) or misalignment (e.g., banner, layout, etc.). The bottleneck lies in this point: Which neurons should be interpreted together? Without principled selection, explanations will be redundant, inconsistent, or even lose prediction relevance. Addressing this selection problem is thus a key step toward advancing MNE.

In this work, we introduce **Neurobasket**, a hierarchical framework that directly addresses the selection and organization problem in MNE. Neurobasket discovers semantic clusters of images via hierarchical clustering, grounds them in natural language through vision–language captioning and LLM

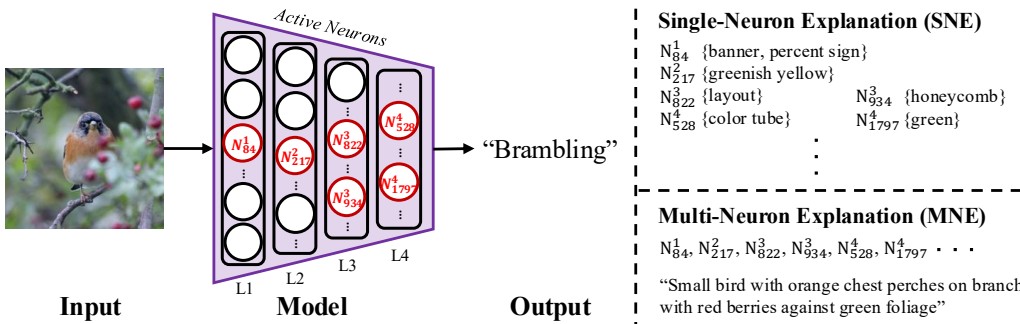

Figure 1: **From SNE to MNE.** Single-Neuron Explanations (SNE) can assign concepts to individual neurons, but this often leads to fragmented or unstable interpretations in distributed representations. To overcome this limitation, Multi-Neuron Explanations (MNE) are proposed, where groups of neurons are interpreted together. The central challenge, however, is identifying which neurons should be grouped to yield coherent and predictive explanations. This selection problem is fundamental for advancing beyond unit-level interpretations. $N_k^l$ denotes $k$-th neuron in layer $l$.

summarization, and identifies neuron sets that are consistently active across cluster features. Each cluster combined with its semantic grounding forms a basket, representing a coherent multi-neuron concept. Beyond discovery, Neurobasket enables set-operation–based analysis (union/difference): unions reveal shared abstractions while differences highlight discriminative features, offering a structured view of how concepts are composed and separated.

We validate Neurobasket through pruning-based causal tests, activation consistency comparisons, and qualitative feature map visualizations. We further demonstrate that Neurobasket generalize across architectures (ResNet and ViT-B/16) and datasets (ImageNet and Places365).

Our contributions are summarized as follows:

- We address the selection problem in Multi-Neuron Explanations by combining consistent activation rules with hierarchical clustering, yielding stable and coherent neuron sets.
- We introduce a hierarchical organization of neuron groups that supports set-operation–based analysis (union/difference), enabling more structured and prediction-aligned interpretations.
- We validate Neurobasket through causal pruning, consistency analysis, and visualizations, and show that it generalizes across diverse architectures and datasets.

Overall, Neurobasket moves beyond non-hierarchical groupings toward explanations that are structurally organized and relationally grounded, offering deeper insights into how complex concepts are encoded within deep neural networks.

## 2 RELATED WORKS

**Neuron-Level Explanations.** Early studies such as NetDissect (Bau et al., 2017) and follow-ups (Zhou et al., 2018; 2021; Wang et al., 2022; Oikarinen & Weng, 2023; Ahn et al., 2024) demonstrated that individual neurons can align with human-interpretable concepts, initiating the study of SNE. While insightful, this perspective is limited in modern architectures, where representations are highly distributed. Prior work has shown that model behavior often depends on activation directions rather than individual units (Morcos et al., 2018), that neurons frequently encode compositional mixtures of concepts (Mu & Andreas, 2020), and that superposition of multiple features within the same neuron is common (Elhage et al., 2022). These findings highlight the need to move beyond explanations based solely on single units. In response, MNE have been proposed, with methods such as FALCON (Kalibhat et al., 2023) and NeurFlow (Cao et al., 2025) demonstrating the value of analyzing neuron groups or activation subspaces. However, most existing approaches define neuron groups in relatively simple or fixed ways, making it difficult to capture how concepts emerge

hierarchically or to analyze relationships such as shared versus discriminative features. Our work extends this direction by introducing a hierarchical framework that constructs **baskets**—multi-neuron groups grounded in semantics and organized across levels of abstraction—thus enabling both coherent interpretation and structured set-operation–based analysis (union/difference).

**Concept directions and sparse-autoencoder features.** Another line of work learns interpretable *directions* in representation space. Concept direction methods (Kim et al., 2018; **?**) train linear classifiers to obtain concept activation vectors in a given layer; these dense directions are effective for detecting the presence of a concept, but each direction mixes many units, which makes it harder to isolate how that concept contributes to the final decision or how it interacts with other neuron groups. Sparse-autoencoder (SAE)–based approaches (Huben et al., 2023; Bussmann et al., 2025) learn a new, sparse latent basis that "unwraps" features into interpretable codes, and Matryoshka SAEs further impose a hierarchy on these latent variables. In all of these methods, the hierarchy or sparsity structure lives in the *learned* representation space, rather than in the original neurons. By contrast, Neurobasket does not learn a new basis: it directly groups concrete neuron indices into baskets (across layers) based on activation statistics over images, and then studies how adding or removing these baskets (union/difference, causal ablation) changes both intermediate semantics and predictions.

**Hierarchical Clustering and Semantic Grounding.** Hierarchical clustering has long been used to reveal multi-level structure in high-dimensional data (Sarfraz et al., 2019; Rousseeuw, 1987; Davies & Bouldin, 2009), while advances in vision–language models (Radford et al., 2021; Bai et al., 2025) have made it possible to associate clusters with human-understandable semantics. Despite this progress, their use in Neuron-level interpretability remains limited: most clustering-based studies operate in image or feature space without explicitly linking semantic abstractions to neuron sets. In contrast, our framework leverages hierarchical clustering to organize image features, grounds them in natural language, and systematically maps these clusters to consistently active neuron groups. This combination enables the discovery of structured, interpretable neuron sets and supports compositional reasoning about how abstract concepts emerge within neural networks.

## 3 NEUROBASKET FRAMEWORK

We propose a framework that organizes neuron responses into baskets, which are multi-neuron groups associated with coherent and human-understandable concepts. The pipeline consists of two stages: (i) basket discovery and semantic grounding, and (ii) neuron set discovery. These steps connect low-level activations with abstract semantic structures, forming the basis for interpreting the model response.

### 3.1 NEUROBASKET DISCOVERY AND SEMANTIC GROUNDING

The goal of this stage is to construct candidate concept baskets that are both semantically coherent and hierarchically organized. We first reveal the hierarchical structure in the activation space through clustering and retain only clusters that are compact, well separated, and semantically consistent. These clusters are then grounded in open-vocabulary language through vision language captioning and LLM summarization, producing human-interpretable baskets.

**Neurobasket Discovery.** For target layer $l$ of the target model $f$, we extract feature representations for all images in the probing set $P$. We then apply FINCH clustering (Sarfraz et al., 2019), which produces a hierarchical organization of the images within that layer. Specifically, features of each target layer are structured into $h$ levels of clusters $\mathcal{C}^{(l)} = \{C_k^{(l,h)}\}$, where $h$ denotes the hierarchy level (from fine-grained to coarse) and $C_k^{(l,h)}$ is the $k$-th cluster at level $h$.

To ensure validity, clusters are filtered according to four criteria: (i) cohesion in feature space, measured by within-cluster compactness and separation (Rousseeuw, 1987), (ii) caption-embedding purity, motivated by prior work on image-text alignment showing that semantic consistency improves clustering quality (Radford et al., 2021), (iii) the Davies–Bouldin index, a standard internal validity score balancing intra-cluster similarity and inter-cluster separation (Davies & Bouldin, 2009), and (iv) the number of valid clusters retained at each hierarchical level, which relates to stability and ro-

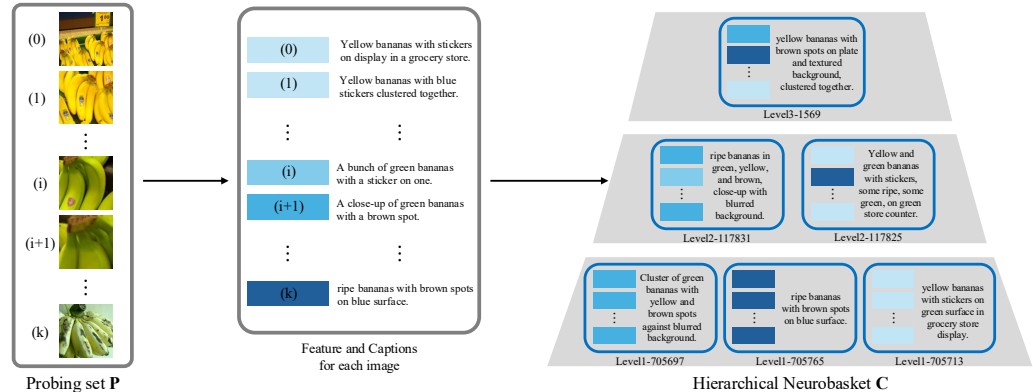

Figure 2: **Overview of basket Discovery and Semantic Grounding.** Neurobasket first applies hierarchical clustering to organize image features into multi-level groups. Then filter groups for semantic consistency. Each cluster is grounded with natural language through vision–language captioning and LLM summarization. Level refers to the hierarchical depth in the clustering tree.

bustness in hierarchical clustering. Clusters and levels that do not satisfy these criteria are discarded. Details of cluster filtering is in Appendix. B.1

**Semantic Grounding.** To associate clusters with human-understandable meaning, each image $x$ is paired with a caption $t(x)$ generated by a vision–language model (Qwen2.5-VL (Bai et al., 2025)). Captions within a cluster are summarized using a large language model (GPT API (Achiam et al., 2023)). A detailed prompt is first applied to obtain a comprehensive description $D_{\text{detail}}$ of the cluster semantics. The simple phrase $D_{\text{simple}}$ is then generated as a concise summary of this detailed description, ensuring consistency between both forms. Details are provided in Appendix. B.10.

By combining each cluster $C_k^{(l,h)}$ with its semantic grounding $(D_{\text{detail}}, D_{\text{simple}})$, we define a basket $B_k^{(l,h)}$. Thus, $B_k^{(l,h)}$ represents a basket that is not only structurally coherent in feature space but also aligned with human-understandable semantics.

## 3.2 NEURON SET CONSTRUCTION

The aim of this stage is to fill each basket with the neurons that consistently activate for its images, so that the basket holds a reliable set of contributing units. Given a basket $B_k^{(l,h)}$, we identify the active neurons for each basket. For each image $x$ and layer $l$, we compute the mean and standard deviation of the activations across all neurons in that layer:

$$\mu^{(l)}(x) = \mathbb{E}[A_N^{(l)}(x)], \quad \sigma^{(l)}(x) = \text{Std}[A_N^{(l)}(x)]. \tag{1}$$

A neuron $N$ at layer $l$ is considered *active* for image $x$ if

$$A_N^{(l)}(x) \geq \mu^{(l)}(x) + \alpha\sigma^{(l)}(x). \tag{2}$$

$\alpha$ denotes the hyperparameter for active neuron selection. (We used $\alpha = 3$ for most cases) Within a basket $= B_k^{(l,h)}$, we then compute the ratio of images in which neuron $n$ is *active*:

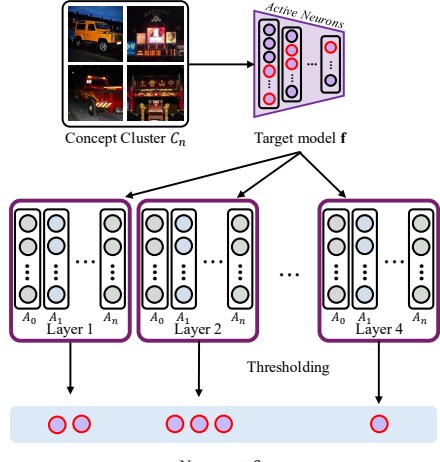

Figure 3: **Overview of Neuron Set Construction.** For each basket, neurons are selected based on consistent activation across its images. Activations exceeding threshold are identified, and only neurons *active* in 75% percent of basket images are retained.

$$p_N^{(l,h)}(B_k^{(l,h)}) = \frac{1}{|B_k^{(l,h)}|} \sum_{x \in B_k^{(l,h)}} \mathbf{1}[N \text{ is active for } x] \tag{3}$$

Table 1: Comparison of activation consistency. We measured average pairwise cosine similarity (higher is better) across baselines. For Neurobasket, 10,000 baskets were randomly sampled. The Random-index baseline selects neurons at a random index, with the same average neuron count as the sampled baskets. Results are averaged over 5 runs with standard deviations reported.

| Metric | Neurobasket (ours) | FALCON | Random-index |
|---|---|---|---|
| **Activation Consistency** | $0.7404 \pm 0.0004$ | $0.5756$ | $0.5542 \pm 0.0250$ |

where $\mathbf{1}[\cdot]$ is the indicator function that equals 1 if the condition is true inside the brackets, and 0 for otherwise.

The neuron set for basket $B_k^{(l,h)}$ is defined as

$$S_k^{(l,h)} = \{u \mid p_N^{(l,h)}(B_k^{(l,h)}) \geq \beta\}. \tag{4}$$

$\beta$ denotes the hyperparameter for neuron set selection (We used $\beta = 0.75$ for most cases). This procedure ensures that a neuron is included in the basket only if it is consistently active in the images belonging to the basket.

## 4 VALIDATING NEUROBASKET VIA ACTIVATION CONSISTENCY

Ideal semantic baskets should share similar representations among the selected set of neurons. To evaluate whether the selected set $S_n$ shares similar representation, we measure activation consistency of $S_n$ across the probing set $P$. Also, we compared Neurobasket with various neuron selection baselines to determine which selection rule (ours vs. baselines) best captures groups with similar representations.

**Mectric.** For a basket $B_n$ and a neuron set $S_n$, let each neuron $(l, j) \in S_n$ have an activation profile $v_{l,j} = [A_{l,j}(x)]_{x \in B_n} \in \mathbb{R}^{|B_n|}$.

An activation consistency is defined as the average pairwise cosine similarity.

$$\text{Consistency}(S_n, B_n) = \frac{2}{|S_n|(|S_n| - 1)} \sum_{\substack{u \neq v \\ u,v \in S_n}} \cos(v_u, v_v). \tag{5}$$

We calculate the average and the standard deviation of $\text{Consistency}$ over baskets.

**Experiment settings.** We use ResNet-50 pretrained on ImageNet as the target model. For each basket, we compute activation consistency using the crop ImageNet validation set, The Random-index baseline (Szegedy et al., 2013) has random sampled the number of neurons same size of the randomly selected baskets. To ensure robustness of the results, we repeat the sampling process five times and report the mean and standard deviation. Details are provided in Appendix B.2.

**Result.** Table 1 shows that Neurobasket achieved the highest activation consistency, indicating that the selected neuron sets capture more coherent representations than the baselines. In particular, Neurobasket showed an average pairwise cosine similarity of 0.7404, substantially higher than FALCON (0.5746) (Kalibhat et al., 2023) and the random-index baseline (0.5542)(Szegedy et al., 2013). These results confirm that our selection rule more effectively groups neurons with shared responses, thereby forming stable and semantically aligned baskets.

## 5 INTERPRETING MODEL RESPONSE VIA NEUROBASKET

### 5.1 UNION AND HIERARCHICAL COMPOSITION

We examine how child baskets combine along the hierarchy and whether their *union* explains more abstract parent concepts.

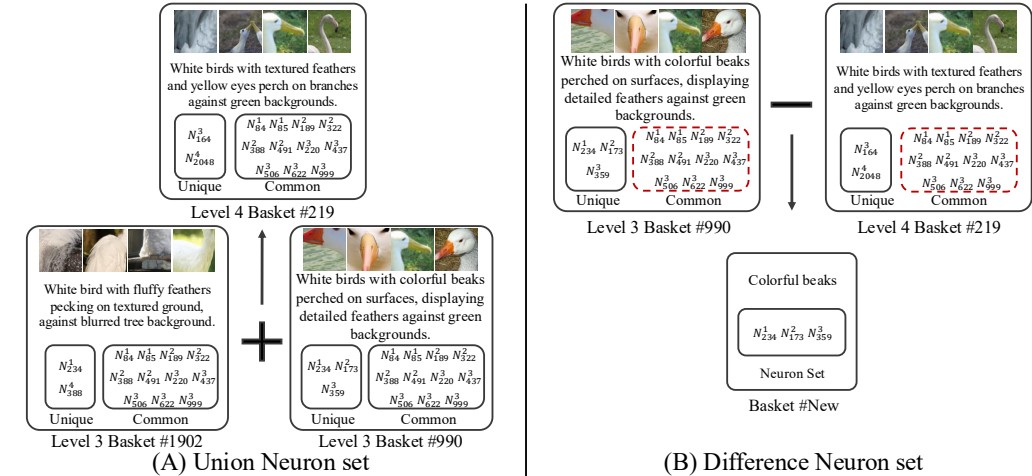

Figure 4: **Union and difference analysis of ResNet-50 baskets.** (Left) The union of sibling baskets captures higher-level abstractions, where fine-grained details (e.g., *feather patterns*, *colorful beaks*) combine into a coherent parent concept (e.g., *white bird with textured feathers and yellow eyes*). (Right) Difference analysis isolates discriminative features by subtr acting shared neurons: removing common cues (e.g., *white bird*, *feather*, *green background*) reveals a more specific concept such as a *colorful beak*. Level refers to the hierarchical depth in the clustering tree.

**Experiment settings.** We analyze ResNet-50 with baskets discovered on crop ImageNet-val. For sibling baskets $B_a^{(h)}$ and $B_b^{(h)}$ merging into $B_p^{(h+1)}$, we study $S_p^{\cup} = S_a \cup S_b$ and compare the parent's textual summary $s_p^{\text{detail}}$ with those of the children. Details are provided in Appendix B.4.

**Result.** Figure 4 (left) illustrates how the union of sibling baskets yields more abstract and coherent parent concepts. For example, detailed child concepts such as *feather patterns* and *colorful beaks* combine into a higher-level abstraction of a *white bird with textured feathers and yellow eyes*. This result highlights that meaningful interpretation does not arise from arbitrary unions of neuron groups, but rather from unions constrained by the hierarchical structure of clustering. By leveraging this hierarchy, Neurobasket provides a structured way to connect fine-grained cues to more abstract semantic categories, thereby supporting faithful interpretation of model representations.

## 5.2 DIFFERENCE AND DISCRIMINATIVE ANALYSIS

We test whether differences between baskets capture basket-specific mechanisms, while common neurons encode shared abstractions.

**Experiment settings.** We analyze ResNet-50 baskets discovered on crop ImageNet-val. For baskets $(B_A, B_B)$ with common neurons, we derive $S_{A \setminus B} = S_A \setminus S_B$. We compare the textual description of A and B, leveraging LLMs (Achiam et al., 2023) to identify the difference. Details are provided in Appendix B.4.

**Result.** Figure 4 (right) demonstrates that analyzing differences between baskets isolates discriminative features that are not discoverable through clustering alone. When common features (e.g., *white bird*, *feather*, and *green background*) are removed, the residual neuron set emphasizes a novel and more specific concept such as a *colorful beak*. This shows that subtraction uncovers basket-specific mechanisms and disentangles overlapping abstractions. Thus, difference analysis complements clustering by revealing additional fine-grained concepts that contribute to the internal representation of the model.

## 5.3 CAUSAL PRUNING EVALUATION

We validate that Neurobasket is able to capture prediction-relevant pathways by measuring the change in class probabilities when pruning concept-aligned baskets.

Table 2: Causal pruning on Resnet-50 ImageNet validation. For class $y$ in imagenet, we prune five baskets whose textual summary mentions $y$ and report the average probability drop $\Delta prob_y$ (larger is better in magnitude).

| Metric | Neurobasket (ours) | Random-basket | Random-index |
|---|---|---|---|
| $\Delta prob_y$ | $-0.1551 \pm 0.0010$ | $-0.0420 \pm 0.0011$ | $-0.0168 \pm 0.0008$ |

**Level4 cluster 3854**

honeycomb pattern with warm golden hues and dark lines on textured surface against warm background.

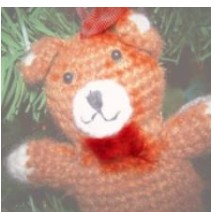 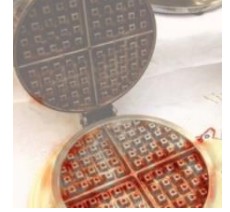

**Level4 cluster 3845**

sharks and fish swim in deep blue water with vibrant background, showing water texture.

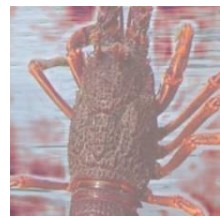 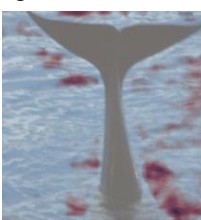

Figure 5: **Qualitative heatmap visualization of ResNet-50 baskets.** Aggregated heatmaps highlight image regions that correspond to the semantic concepts identified in textual summaries (e.g., honeycomb patterns, bright floral textures in blue, green, and yellow). Level refers to the hierarchical depth in the clustering tree.

**Experiment settings and Metrics.** We use ResNet-50 as the target model. baskets are constructed from crop Imagenet set. For matched ImageNet class $y$, we randomly select 5 baskets whose textual summary contains the class name and ablate $S_n$, masked at inference on the class's validation images. We report the average probability drop. $\Delta prob_y = \frac{1}{|X_y|} \sum_{x \in X_y} \left( prob(y \mid x) - p^{\text{pruned}}(y \mid x) \right)$. Baselines include random-index neurons (size-matched) Szegedy et al. (2013), random baskets. To ensure robustness of the results, we repeat the sampling process five times and report the mean and standard deviation. Details are provided in Appendix B.5.

**Results.** Table 2 shows the effect of ablating concept-aligned baskets on class probabilities. We observe that pruning our discovered baskets shows a substantial average drop in the target class probability ($\Delta prob_y = -0.1558$), whereas pruning random baskets ($-0.0429$) or random-index neurons ($-0.0168$) results in only minor changes. This clear gap indicates that Neurobasket capture features closely coupled to class identity, while random groupings rarely disrupt predictions. Overall, the results support our claim that Neurobasket can represent causally relevant pathways for model decisions rather than arbitrary co-activations.

## 5.4 FEATURE MAP VISUALIZATION

We qualitatively evaluate whether Neurobasket localizes semantically coherent regions that are consistent with their textual summaries.

**Implementation details.** We visualize highlighted region of a basket from ResNet-50 by upsampling activation maps and aggregating group masks. Details are in appendix B.6.

**Result.** Figure 5 shows that baskets reliably highlight spatial regions that align with their semantic summaries. For example, baskets described by *honeycomb patterns* or *vibrant background, showing water texture* exhibit activations align precisely on those textures and colors in the image. Such localization indicates that the grouped neurons do not respond arbitrarily, but instead consistently capture semantically coherent cues. These qualitative results support our claim that Neurobasket provide faithful grounding of neuron groups, where the highlighted regions closely match the text-level concepts assigned to each basket, thereby strengthening the interpretability of the underlying model representations.

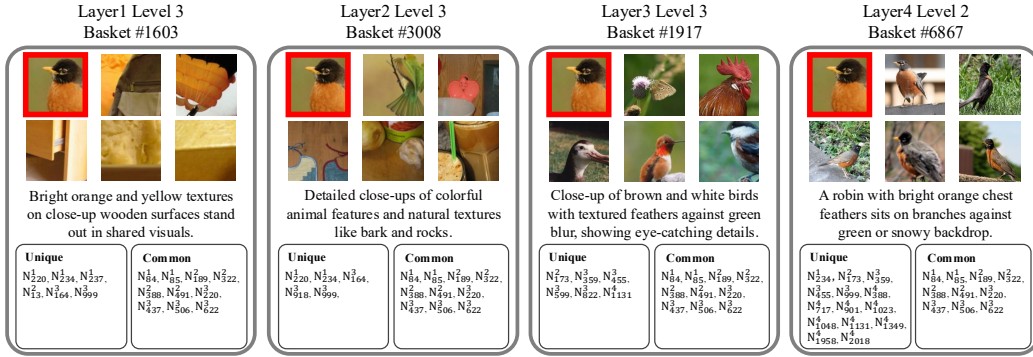

Figure 6: **Qualitative visualizations across layers in ResNet-50.** Baskets constructed from earlier layers emphasize low-level features such as textures, colors, or simple shapes. In contrast, deeper layers give rise to baskets that capture high-level and abstract concepts, highlighting more semantically complex regions. This demonstrates how neuron selection progressively shifts from low-level detail to abstract representations as depth increases. Level refers to the hierarchical depth in the clustering tree.

## 5.5 ANALYSIS OF BASKET FROM VARIOUS LAYERS.

We investigate how baskets constructed from different layers capture concepts at varying levels of abstraction. Earlier layers are expected to focus on low-level cues such as edges, textures, and simple patterns, whereas later layers should emphasize higher-level and more abstract semantic categories. Details are provided in Appendix B.7.

**Result.** Figure 6 confirms that baskets derived from different depths highlight semantically distinct levels of abstraction. For common input images, baskets from shallow layers tend to emphasize localized and low-level cues such as color patches or repetitive textures, reflecting the primitive features encoded at those stages. Baskets formed from these early-layer features primarily select neurons close to the input, supporting the representation of low-level concepts like colors and textures. In contrast, baskets constructed from deeper-layer activations capture more abstract and holistic concepts (e.g., object parts or composite patterns). These baskets tend to recruit neurons located closer to the final layers of the network, reflecting the model's change in representation toward higher-level semantics. In general, this pattern aligns with the known hierarchy of deep neural networks (Bau et al., 2017; Mu & Andreas, 2020; Kalibhat et al., 2023), showing that Neurobasket not only recovers semantic abstraction across depth, but also reveals how neuron selection shifts spatially from front-layer to last-layer units as clustering is performed in deeper layers. Additional qualitative results are provided in Appendix. A.2

## 6 GENERALIZATION AND ABLATION STUDIES

### 6.1 VIT ARCHITECTURES (IMAGENET-PRETRAINED)

We examine whether the proposed framework extends beyond convolutional networks to transformer architectures. In particular, we evaluate activation consistency, causal pruning effects, and qualitative figures on ViT-B/16 to show whether NeuroBasket discovers coherent and prediction-relevant neuron groups in attention-based models.

**Implementation details.** We use ViT-B/16 pretrained on ImageNet-1k with crop ImageNet-val as $\mathcal{P}$. We report $\text{Consistency}(S_n, C_n)$ across depth, average $\Delta prob_y$ from pruning. Details are provided in Appendix B.8.

**Result.** Our analysis confirms that the Neurobasket framework also works on to transformer architectures. Table 3 shows that on ViT-B/16, Neurobasket achieves very high activation consistency (0.9331), whereas the Random-index baseline produces near-zero similarity (0.0092). This indicates

Table 3: Activation consistency in ViT-B/16. We measured average Activation Consistency (higher is better). For Neurobasket, 10,000 baskets were randomly sampled. The Random-index baseline selects neurons at a random index, with the same average neuron count as the sampled baskets. Results are averaged over 5 runs with standard deviations reported. $\alpha$ denotes the threshold constant for identifying active neurons.

| Metric | Neurobasket (ours) | Random-index |
|---|---|---|
| **Activation Consistency** | $0.9320_{\pm0.0008}$ | $0.0092_{\pm0.0190}$ |

Table 4: Causal pruning on ViT-B/16 ImageNet validation. For class $y$ in imagenet, we prune baskets whose textual summary mentions $y$ and report the average probability drop $\Delta prob_y$ (larger is better in magnitude).

| Metric | Neurobasket (ours) | Random-basket | Random-index |
|---|---|---|---|
| $\Delta prob_y$ | $-0.2345_{\pm0.0005}$ | $-0.2237_{\pm0.0009}$ | $-0.0047_{\pm0.0003}$ |

that even in attention-based models, Neurobasket identifies neuron groups with strongly aligned activation profiles, while random selection fails to recover coherent sets.

Causal pruning results in Table 4 further validate the causal relevance of the discovered baskets. When pruning baskets whose textual summaries mention the target class, Neurobasket shows a substantial probability drop (-0.2344), than random-basket pruning (-0.2237) and the Random-index baseline (-0.0046). These results demonstrate that the selected neuron groups are not only internally consistent but also causally tied to model predictions.

Finally, the qualitative results in Figure A.8 illustrate also shows similar trend in Sec. 5.5. Additional qualitative results are provided in Appendix. A.1 and A.2.

Together, these results demonstrate that Neurobasket generalizes beyond convolutional backbones, yielding coherent and causally meaningful interpretations in transformer-based vision models.

## 6.2 RESNET-18 (PLACES365-PRETRAINED)

We evaluate Neurobasket generalizes not only across architectures but also across different pretraining datasets. In particular, this experiment tests robustness to the choice of probing set, verifying that our method still yields coherent baskets even when evaluated on Places365.

**Implementation details.** We use ResNet-18 pretrained on Places365 and construct baskets from its activations. For the probing set $P$, we follow the same cropping protocol as in Sec. 4, but apply it to the Places365 validation set instead of ImageNet. This allows us to assess whether the activation consistency trends observed on ImageNet also hold when both the architecture and probing distribution are changed. Details are provided in Appendix B.9.

**Result.** Table 5 shows that Neurobasket maintains a clear advantage over the Random-index baseline on ResNet-18 pretrained with Places365. Our method achieves an average pairwise similarity of 0.7047, substantially higher than the baseline value of 0.5699. This demonstrates that the framework generalizes beyond both ImageNet-trained models and standard ResNet-50 backbones, consistently identifying coherent neuron groups under changes in architecture and pretraining distribution.

Furthermore, qualitative figure in Figure A.7 illustrate also shows similar trend in Sec. 5.5.Additional qualitative results are provided in Appendix. A.1 and A.2. Together, these results confirm that Neurobasket not only preserves the hierarchical evolution of concepts across network depth but also sustains consistency across diverse datasets.

## 7 CONCLUSION

We introduced **Neurobasket**, a hierarchical framework for Multi-Neuron Explanations that addresses the selection and organization problem by leveraging hierarchical clustering. Neurobas-

Table 5: Activation consistency in ResNet-18 pretrained in Places365 dataset. We measured average Activation Consistency (higher is better). For Neurobasket, 10,000 baskets were randomly sampled. The Random-index baseline selects neurons at a random index, with the same average neuron count as the sampled baskets. Results are averaged over 5 runs with standard deviations reported. $\alpha$ denotes the threshold constant for identifying active neurons.

| Metric | Neurobasket (ours) | Random-index |
|---|---|---|
| **Activation Consistency** | $0.7041_{\pm 0.0003}$ | $0.5699_{\pm 0.0343}$ |

ket discovers coherent neuron sets aligned with model predictions and grounds them in semantics, enabling structured reasoning such as union, intersection, and difference. Our evaluations with pruning-based tests, activation consistency, and qualitative visualizations show that baskets capture meaningful and prediction-relevant concepts, while generalizing across architectures and datasets. This work advances beyond unit-centric or non-hierarchical approaches, offering a more faithful and systematic view of neural representations.

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

## A  ADDITIONAL QUALITATIVE RESULTS

### A.1  ADDITIONAL UNION AND DIFFERENCE ANALYSIS

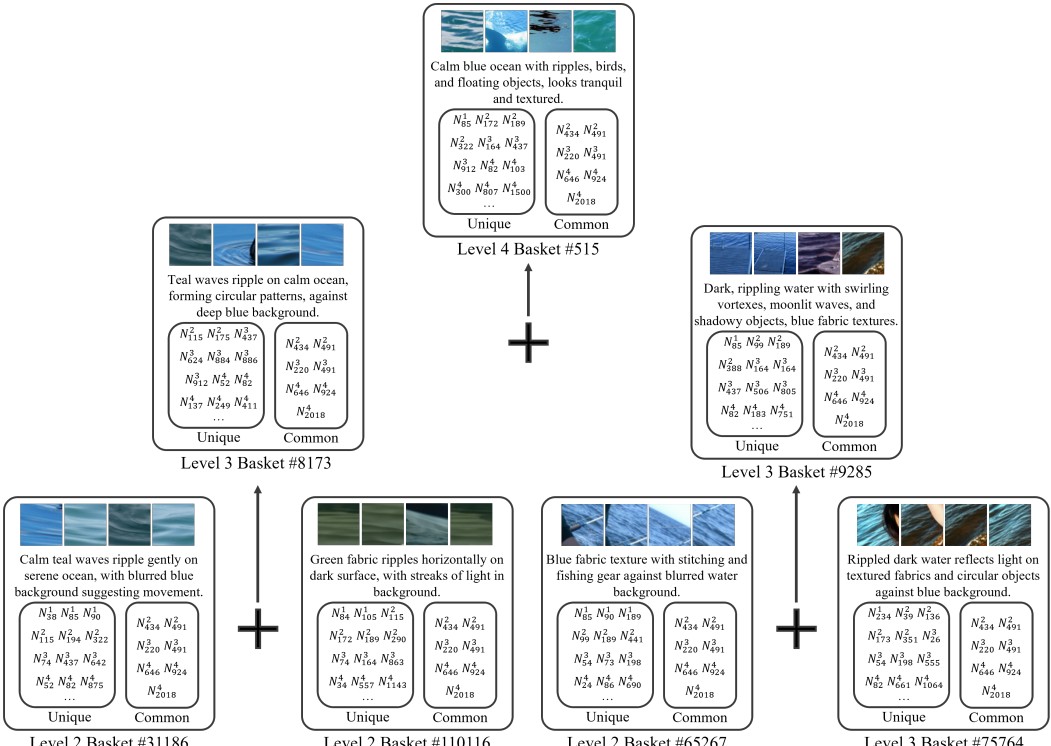

Figure A.1: **3-Level analysis of ResNet-50 baskets** Union operations combine sibling baskets into more abstract parent concepts along hierarchical level h.

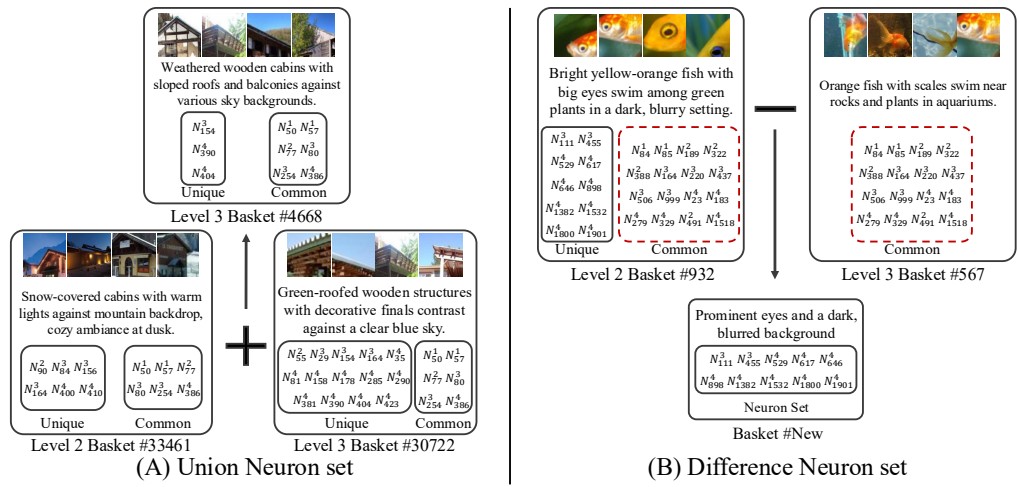

Figure A.2: **Union and difference analysis of ResNet-18 baskets.** Union operations combine sibling baskets into more abstract parent concepts, while difference analysis highlights basket-specific cues (e.g., fine-grained shape or abstract concepts) that clustering alone may not disentangle.

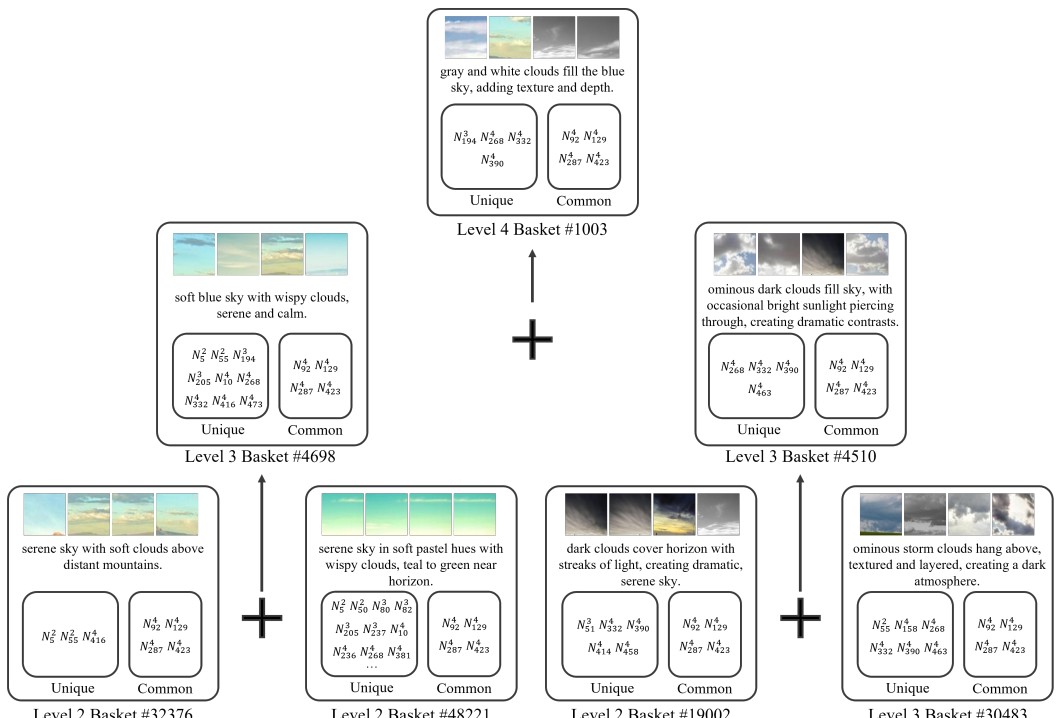

Figure A.3: **3-Level analysis of ResNet-18 baskets** Union operations combine sibling baskets into more abstract parent concepts along hierarchical level h.

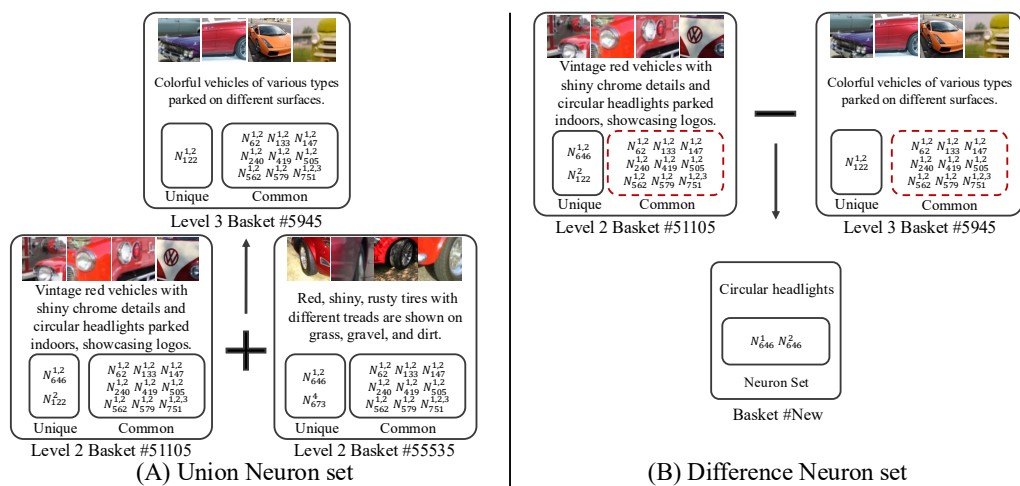

(A) Union Neuron set        (B) Difference Neuron set

Figure A.4: **Union and difference analysis of ViT-B/16 baskets.** Union operations combine sibling baskets into more abstract parent concepts, while difference analysis highlights basket-specific cues (e.g., fine-grained shape or abstract concepts) that clustering alone may not disentangle.

**Result.** These results further confirm that Neurobasket supports structured set-theoretic reasoning across architectures, where union represents more abstract concept as combination of child representations. Also difference isolates discriminative semantics which hard disentangled by clustering.

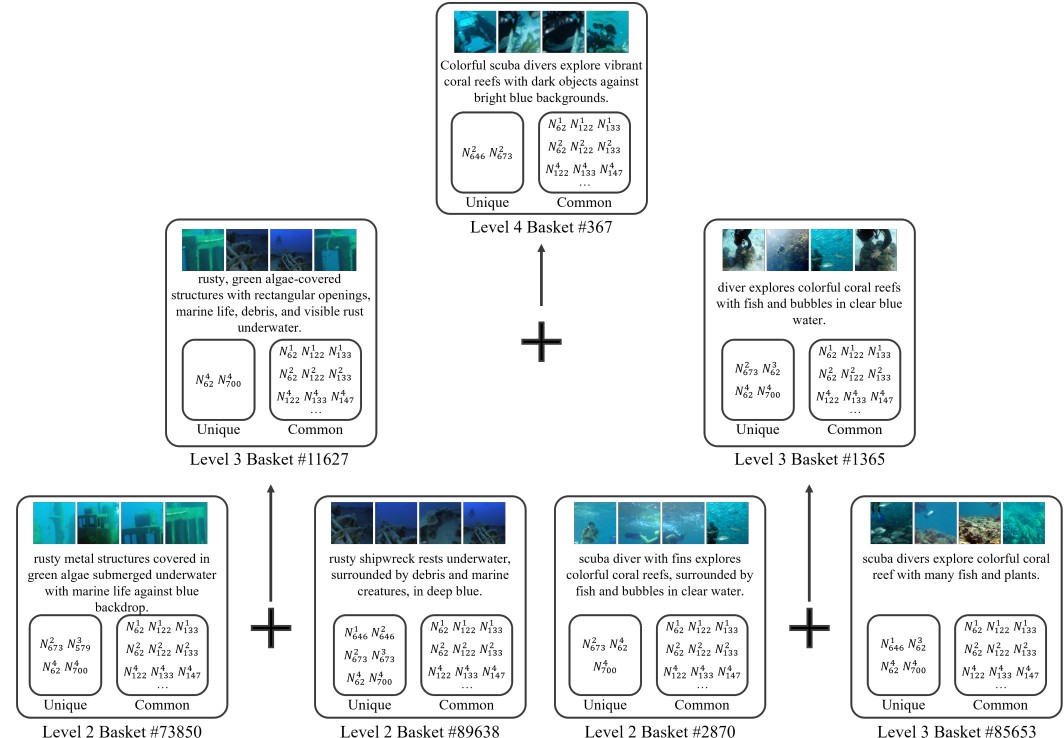

Figure A.5: **3-Level analysis of ViT/B-16 baskets** Union operations combine sibling baskets into more abstract parent concepts along hierarchical level h.

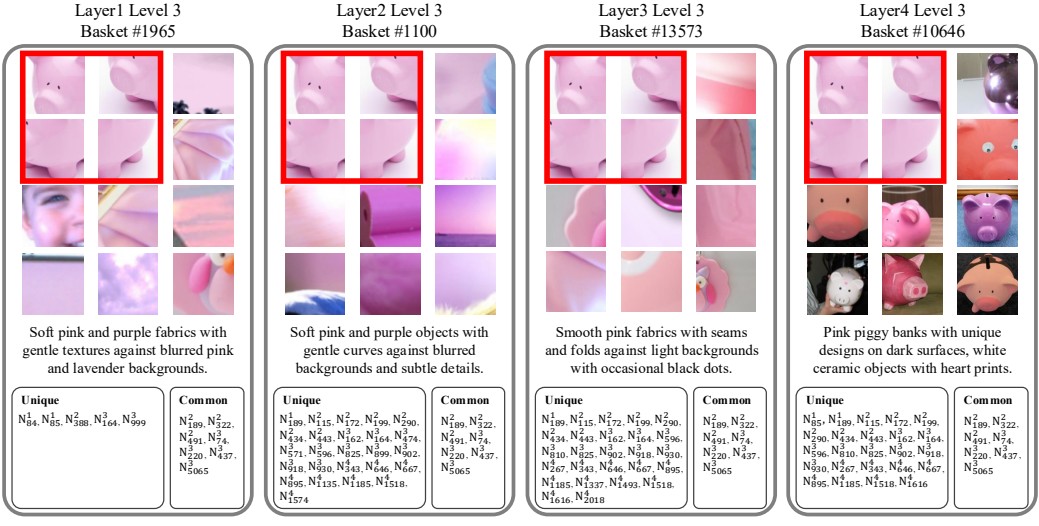

Figure A.6: **Qualitative visualizations across layers in ResNet-50.** Shallow layers focus on low-level cues such as edges, textures, or colors, while deeper layers capture higher-level semantic regions, consistent with hierarchical concept abstraction.

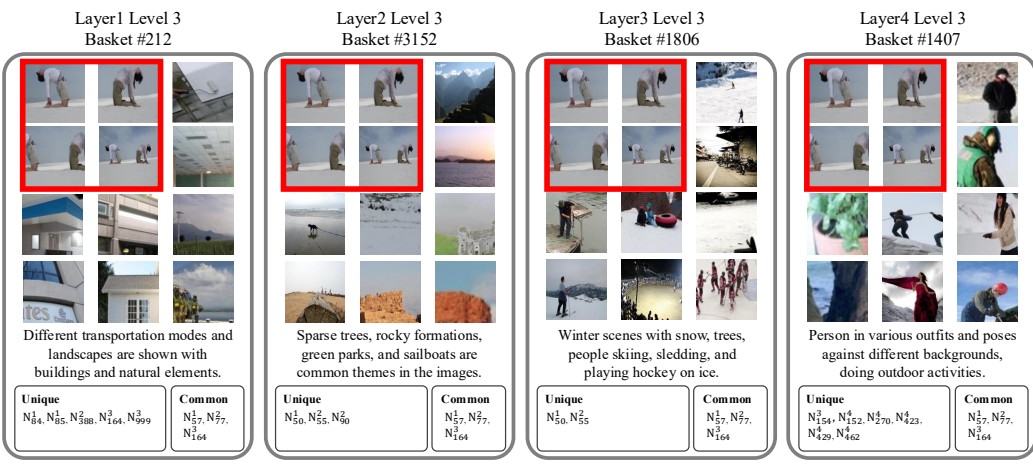

Figure A.7: **Qualitative visualizations across layers in ResNet-18.** Shallow layers focus on low-level cues such as edges, textures, or colors, while deeper layers capture higher-level semantic regions, consistent with hierarchical concept abstraction.

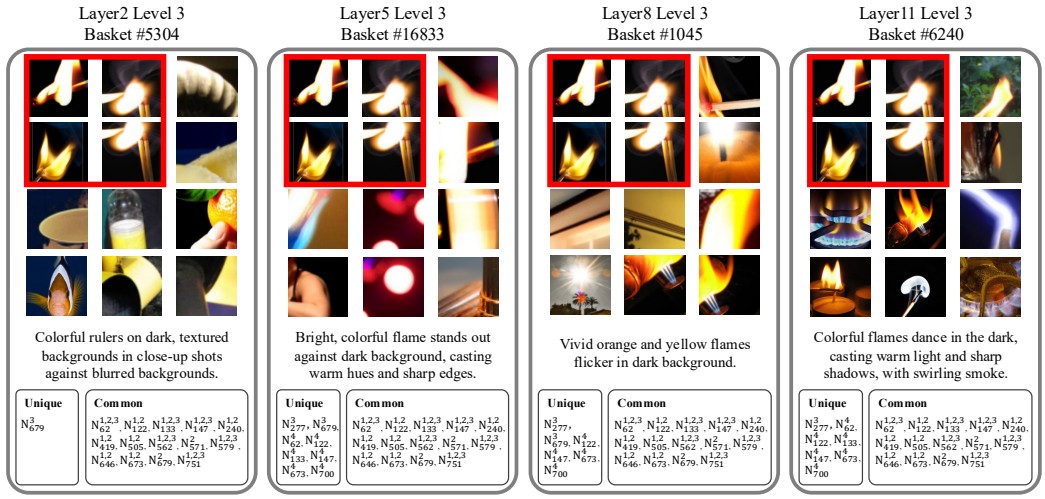

Figure A.8: **Qualitative visualizations across layers in ViT-B/16.** Shallow layers focus on low-level cues such as edges, textures, or colors, while deeper layers capture higher-level semantic regions, consistent with hierarchical concept abstraction.

## A.2 ADDITIONAL LAYER-WISE QUALITATIVE RESULTS

**Result.** Together, these qualitative results illustrate that Neurobasket consistently capture hierarchical concept evolution across both convolutional and transformer models, confirming the framework's generality beyond single architectures or datasets.

## B EXPERIMENT SETTINGS AND IMPLEMENTATION DETAILS

### B.1 BASKET FILTERING ANALYSIS

**Model and Data.** All baskets used across *all* experiments are generated under a single, consistent protocol described here.

**Clustering.** We apply FINCH (Sarfraz et al., 2019) to activations taken at the end of each layer block of the target model (e.g., ResNet-50 `layer1.2`, `layer2.3`, `layer3.5`, `layer4.2`). The exact layer selections used by each experiment are specified in the corresponding subsections of this appendix.

**Filtering metrics.**

- *Cohesion* inspired from the silhouette coefficient (Rousseeuw, 1987).
- *Purity* as the average cosine similarity between CLIP (Radford et al., 2021) image embeddings and the cluster-level caption embedding.
- *Davies–Bouldin index (DBI)* (Davies & Bouldin, 2009) for intra- vs. inter-cluster separation. Due to the computation limitation, we used simplified version of DBI index.

**Filtering procedure.** For each metric, clusters with scores exceeding $\mu + 2\sigma$ are treated as outliers and removed, eliminating roughly $5\%$ of clusters per criterion on average. Any cluster failing *at least one* criterion is discarded. For better interpretation, we decided not to analyze higher hierarchy levels when the number of clusters at a level becomes comparable to or smaller than the number of dataset classes (typically beyond level 5).

**Reproducibility notes.** Unless otherwise stated, all ImageNet experiments use the cropped ImageNet validation set (center-crop $224\times224$ plus grid crops at scales $\{56, 112, 224\}$; $\approx$2.95M images) as the probing set $\mathcal{P}$.

## B.2 IMPLEMENTATION DETAILS FOR ACTIVATION CONSISTENCY (SEC. 4)

**Model and Data.** We use ResNet-50 pretrained on ImageNet (PyTorch official weights). The probing set $\mathcal{P}$ is built from the ImageNet validation set with center-crop $224\times224$ and additional grid crops at scales $\{56, 112, 224\}$, totaling $\approx$2.95M images. Unless noted, activations are extracted from the end of `layer4`.

**Clustering.** FINCH is applied to `layer4` activations, producing a 4-level hierarchy.

**Evaluation settings.**

- *Neurobasket (ours).* We randomly sample 10,000 baskets across hierarchy levels and compute activation consistency as the mean pairwise cosine similarity between neuron activation profiles within each basket, followed by averaging across baskets.
- *FALCON* (Kalibhat et al., 2023). Using `layer4` features on the cropped ImageNet validation set, we follow the thresholds reported in the original paper to obtain 344 neuron groups and report their mean activation consistency.
- *Random-index baseline* (Szegedy et al., 2013). For each basket, we match the average neuron count and compute activation consistency on randomly selected unit indices.

Evaluations involving random sampling (ours and random-index) are repeated five times with different seeds; we report the mean and standard deviation.

## B.3 ANALYSIS OF THE ACTIVE-NEURON THRESHOLD $\alpha$ AND $\beta$

**Neuron selection.** A neuron $N$ in layer $l$ is considered *active* for image $x$ if

$$A_N^{(l)}(x) \ \geq \ \mu^{(l)}(x) \ + \ \alpha \cdot \sigma^{(l)}(x),$$

with $\alpha \in \{1, 2, 3, 5, 7\}$.

**Basket construction.** For each basket, neurons active in at least $75\%$ of its images ($\beta$=0.75) are retained in the neuron set $S_n$.

**Evaluation.** For each $\alpha$, we randomly sample 10,000 baskets discovered from `layer4` features of ResNet-50 and compute activation consistency (mean off-diagonal cosine similarity across neuron profiles), aggregating results over baskets..

Table A.1: Sensitivity of basket construction to the activation threshold $\alpha$ (ResNet-50 layer4, $\beta = 0.75$). Baseline top-1 accuracy without ablation is $0.8565$ for all settings.

| $\alpha$ | Mean #neurons / basket | Mean consistency (off-diag cos) | Mean $\Delta\text{logit}_{gt}$ / image |
|------|------|------|------|
| 2.0 | 37.19 | 0.5863 | 0.5464 |
| 3.0 | 18.52 | 0.6261 | 0.3504 |
| 5.0 | 6.25 | 0.7002 | 0.1335 |
| 7.0 | 3.15 | 0.7067 | 0.1206 |

Table A.2: Sensitivity of basket construction to the activity ratio threshold $\beta$ (ResNet-50 layer4, $\alpha = 3.0$). Baseline top-1 accuracy without ablation is $0.8565$ for all settings.

| $\beta$ | Mean #neurons / basket | Mean consistency (off-diag cos) | Mean $\Delta\text{logit}_{gt}$ / image |
|------|------|------|------|
| 0.50 | 31.49 | 0.5875 | 0.5576 |
| 0.70 | 20.03 | 0.6196 | 0.4055 |
| 0.75 | 18.52 | 0.6261 | 0.3504 |
| 0.80 | 17.25 | 0.6331 | 0.2735 |
| 0.90 | 11.35 | 0.6669 | 0.1920 |
| 1.00 | 10.40 | 0.6755 | 0.1265 |

**Results.** We evaluated hyperparameter sensitivity on the neuron-selection thresholds $\alpha$ and $\beta$. As $\alpha/\beta$ increase, baskets become smaller and more internally consistent (off–diagonal cosine similarity increases), but the mean $\Delta\text{logit}_{gt}$/image under ablation decreases—reflecting the expected trade–off between selecting highly coherent units and selecting units with maximal causal impact. Our goal is to identify neuron sets that are both stable and decision–relevant; the ablation shows that Neurobasket behaves smoothly across this range, and that our default ($\alpha{=}3.0, \beta{=}0.75$) is a balanced operating point where we obtain strongly consistent baskets while still inducing substantial, structured changes in the model's predictions.

## B.4 IMPLEMENTATION DETAILS FOR UNION AND DIFFERENCE ANALYSES (SEC. 5.1, SEC. 5.2)

**Setup.** We analyze baskets discovered from ResNet-50 `layer4` with the cropped ImageNet validation set.

**Union (hierarchical composition).** For sibling baskets $B_a^{(h)}$ and $B_b^{(h)}$ merged into parent $B_p^{(h+1)}$, we form $S_p^\cup = S_a \cup S_b$ and compare the parent's textual summary to those of its children to assess abstraction.

**Difference (discriminative analysis).** For a pair $(B_A, B_B)$ with overlap, we compute $S_{A \setminus B} = S_A \setminus S_B$. Cluster captions and residual differences are summarized using a GPT API; we qualitatively verify that subtracting shared neurons surfaces basket-specific semantics.

## B.5 IMPLEMENTATION DETAILS FOR CAUSAL PRUNING (SEC. 5.3)

**Class–basket matching.** We match ImageNet class names to basket captions with word-boundary–aware regex (case-insensitive), forming (class $y$) $\leftrightarrow$ {(level, basket)} pairs. If a class has many matches, we randomly subsample a fixed number per class for balance (seeded for reproducibility).

**Neuron ablation.** At inference, we ablate basket units via forward hooks that set the targeted channels to zero (*scale* $= 0$). For 4D tensors (conv blocks) we zero `[batch, channels, :, :]`; for 2D tensors we zero `[batch, units]`; for 3D tensors we zero only the CLS token positions by default.

**Metric.** For class $y$, we report the average probability drop

$$\Delta\text{prob}_y = \frac{1}{|X_y|} \sum_{x \in X_y} \left( \text{prob}(y \mid x) - \text{prob}^{\text{pruned}}(y \mid x) \right).$$

**Baselines and repeats.** We compare against (i) *Random-index* (size-matched random units across the same module set) and (ii) *Random-basket* (class-wise number of baskets matched to ours, sampled from the valid basket pool). Experiments involving random sampling are repeated five times with different seeds; we report mean $\pm$ std.

### B.6    IMPLEMENTATION DETAILS FOR FEATURE MAP VISUALIZATION (SEC. 5.4)

**Layers and data.** We visualize representative baskets from the end of each ResNet-50 layer block(e.g., `layer1.2`, `layer2.3`, `layer3.5`, `layer4.2`), using the cropped ImageNet validation set.

**Procedure.** For a basket $S_n$, per-unit activation maps are upsampled to $224\times224$, normalized to $[0, 1]$ (percentile-based normalization with $p_{\text{lo}}=1$, $p_{\text{hi}}=99$), and averaged across units. Overlays are produced by blending the heatmap with the original image (resized to 256 and center-cropped to 224) using a white-to-red colormap with $\alpha=0.5$. We report all-layers overlay produced by averaging all-neuron heatmaps apply top-percent masking (maintain only top 30%).

### B.7    IMPLEMENTATION DETAILS FOR LAYER-WISE CONCEPT EVOLUTION (SEC. 5.5)

We visualize baskets discovered from different depths of the network to investigate how concepts evolve across layers. Following the same probing set and clustering procedures described in Sec. 5.5, we extract activations from multiple layer blocks (e.g., ResNet-50 layer1.2, layer2.3, layer3.5, layer4.2) and construct hierarchical baskets. For each basket, consistently active neurons are identified using $\alpha = 3$ and $\beta = 0.75$.

### B.8    IMPLEMENTATION DETAILS FOR VIT-B/16 (SEC. 6.1)

**Model.** We use `torchvision` ViT-B/16 (ImageNet-1k pretrained; 12 transformer blocks, hidden size 768, patch size 16).

**Layer mapping and hooks.** For analyses that address block-level features, we follow the standard mapping layer$\{1, 2, 3, 4\} \rightarrow$ block$\{2, 5, 8, 11\}$. When ablating 3D tensors, we zero only CLS-token channels by default.

**Data and protocol.** We use the cropped ImageNet validation set as $\mathcal{P}$. Activation consistency and pruning are computed analogously to the ResNet-50 setting, with basket discovery performed on ViT features and identical sampling/repeat protocols.

### B.9    IMPLEMENTATION DETAILS FOR RESNET-18 (PLACES365) (SEC. 6.2)

**Model.** We evaluate ResNet-18 pretrained on Places365 (365 classes); checkpoints are loaded into a standard `torchvision` ResNet-18 head. We used same checkpoint from Ahn et al. (2024)

**Categories and data.** We construct $\mathcal{P}$ from the Places365 validation set using the same cropping protocol as for ImageNet.

**Protocol.** Activation consistency, follow the same procedures as in the ResNet-50 setting, with layer-block endpoints serving as hook locations.

**Environment and Defaults.** Unless otherwise stated: PyTorch and `torchvision` official weights are used; preprocessing follows the standard ImageNet mean/std. All reported results with random sampling are averaged over 5 seeds with mean$\pm$std.

### B.10 USE OF LLMS

Large language models (LLMs) were used for *semantic grounding* of discovered neuron groups. Specifically, we employed a vision–language model (Qwen2.5-VL) to generate raw captions for each cluster, and then applied an instruction-tuned LLM (GPT-3.5 turbo API) to summarize and refine these captions into concise textual descriptions. Prompts consisted of plain English instructions requesting short, human-readable summaries without stylistic emphasis. No LLMs were used for data preprocessing, clustering, filtering, or evaluation; all experimental results (activation consistency, pruning, visualization, etc.) are based exclusively on deterministic computations implemented in PyTorch. We provide representative prompt templates and outputs below to ensure reproducibility and transparency. In addition, ChatGPT-5 was used to polish the writing of the manuscript. All sentences in the submitted paper were reviewed, edited, and approved by the authors to ensure accuracy and clarity.

**Example of prompt for $D_{detail}$**

```
system_prompt = """
    You are a captioning assistant that extracts the most specific shared
    ↪  visual identity across multiple image captions for use in a T2I
    ↪  prompt.

    Inclusion rules:
    - A feature is included only if it appears in  ceil(0.75 * N)
    ↪  captions after normalization (e.g., N=8  6).
    - This applies to all categories: subject, position, shape, color,
    ↪  texture, composition, clarity, count, background.
    - Normalize synonyms (e.g., "plate" = "dish", "center" = "centered")
    ↪  before counting.
    - Treat multiple sentences in one caption as describing the same
    ↪  image.

    Description rules:
    - Only describe features that meet the threshold.
    - Prioritize low-level features: shape, surface, texture, layout,
    ↪  position.
    - Include a central subject only if it meets the threshold.
    - Do not include any subject or object as the central concept unless
    ↪  it appears in at least ceil(0.75 * N) captions. This includes
    ↪  avoiding even seemingly prominent or salient entities such as
    ↪  animals or people if they do not meet the threshold.
    - If no subject meets the threshold, do not include any subject at
    ↪  allnot even general ones (e.g., "animal", "creature",
    ↪  "figure")and instead focus entirely on qualifying low-level
    ↪  features.
    - If no single subject/concept appears in at least ceil(0.75 * N)
    ↪  captions, do not mention any subjects, categories, or types of
    ↪  objectseven in vague or plural form. Do not list or summarize
    ↪  them. Focus only on qualifying low-level visual features.
    - Avoid vague terms such as "some", "various", or category-level
    ↪  generalizations like "textured surface". Use specific terms only
    ↪  if they meet the threshold; otherwise omit.
    - Each concept appears only once.
    - Do not use vague texture terms like "textured surface" or "various
    ↪  textures". Describe texture only if a specific texture (e.g.,
    ↪  rough, smooth, glossy) appears in  ceil(0.75 * N) captions after
    ↪  normalization. If no single texture meets the threshold, omit
    ↪  texture entirely.

    Output constraints:
    - Max 50 words, max 4 comma-separated phrases.
    - Rank by frequency and include: 1) subject, 2)
    ↪  shape/color/surface/texture, 3) position, 4) background, 5)
    ↪  composition.
    - If too long, prune in reverse order.
```

```
    - Output one declarative sentence, in third-person present tense, no
    ↪  meta-phrases, no "image/photo".
    """

    user_prompt = f"""
    Each bullet below is a caption describing one image. Captions may
    ↪  contain multiple sentences.

    {chr(10).join('- ' + t for t in texts)}

    Summarize the shared visual identity under the 75% rule. Follow the
    ↪  system rules.
    """
```

**Example of prompt for $D_{simple}$**

```
    system_prompt = (
        "You rewrite technical or detailed descriptions into one short,
        ↪  plain-English sentence. "
        "Focus on these, if present: 1) subject, 2)
        ↪  shape/color/surface/texture, 3) position, 4) background, 5)
        ↪  composition. "
        "Rules: max 15 words; no commas; no meta-phrases like
        ↪  'image/photo';"
        "single sentence ending with a period; capture the core subject
        ↪  and one key qualifier."
    )
    user_prompt = f"Rewrite this into one very short, simple sentence for
    ↪  a human:\n\n{long_sentence}"
```

**Example of prompt for difference analysis**

```
    system_prompt = (
    "Compare two sentences and extract the **key concept unique to
    ↪  sentence 1 only**.\n"
    "Output **only one line**: a **keyword or a short sentence**.\n"
    "Do **not** include anything from sentence 2.\n"
    "No explanations, no extra text | just the one-line result."
)

user_prompt = (
    f"1) {sentence_1}\n"
    f"2) {sentence_2}\n\n"
    "Give only the concept unique to sentence 1. Output just one line."
)
```

### B.11 REPRODUCIBILITY

To ensure reproducibility, we will release the full implementation of *Neurobasket*, including experiment scripts, evaluation protocols, and prompt templates, under an open-source license later.

## C TEXT–IMAGE SIMILARITY ANALYSIS (SENTENCEBERT AND CLIP)

To quantitatively assess the semantic faithfulness and interpretability of basket captions beyond activation-level consistency, we additionally measure how well basket summaries align with the underlying image captions and visual features using SentenceBERT and CLIP.

### C.1 SENTENCEBERT SIMILARITY.

**Experiment settings.** We first sample 10,000 baskets across hierarchy levels. For each basket, we embed (i) its summarized basket caption and (ii) all image captions that were used to construct and summarize this basket (before LLM refinement) using SentenceBERT. We then compute the cosine

similarity between the basket caption and each of these image captions and average over them to obtain a per-basket similarity score. As a random baseline, for each basket, we randomly sample the same number of image captions from the global caption pool (i.e., captions belonging to other baskets) and compute the average SentenceBERT similarity to the basket caption.

**Results.** Averaging over 10,000 baskets yields

$$\text{Neurobasket} : \mu = 0.5623, \sigma = 0.1121 \quad \text{vs.} \quad \text{Random} : \mu = 0.2238, \sigma = 0.0645,$$

showing that basket captions are substantially more aligned with their own supporting image captions than with unrelated captions.

## C.2 CLIP SIMILARITY.

We also evaluate text–image alignment directly in the model's representation space.

**Experiment settings.** We randomly sample 10,000 baskets discovered from the final convolutional block (ResNet-50 `layer4`) and collect their representative images (the same images used to define each basket). For each basket, we encode the images with the CLIP image encoder and the basket caption with the CLIP text encoder, and compute the average cosine similarity between the image features and the caption feature. As a random baseline, we keep the same set of images but pair them with randomly selected captions from other baskets.

**Results.**

$$\text{Neurobasket} : \mu = 0.2650, \sigma = 0.0250 \quad \text{vs.} \quad \text{Random} : \mu = 0.2020, \sigma = 0.0140.$$

These results indicate that Neurobasket's textual summaries are not only consistent with the underlying image captions but also well aligned with the model's visual feature space, supporting the faithfulness of the proposed semantic descriptions.

## D PARENT–CHILD BASKET CONSISTENCY ANALYSIS

To verify that the discovered hierarchy behaves in a set-theoretic manner, we additionally analyze how well each parent basket is explained by the union of its children.

**Experiment Settings.** From all valid level-1 $\rightarrow$ level-2 parent–child pairs in ResNet-50 `layer4`, we randomly sample 50 parent baskets and, for each parent $P$ with children $\{C_k\}$, compute

$$U = \bigcup_k C_k, \quad \text{parent\_coverage} = \frac{|P \cap U|}{|P|}, \quad \text{union\_coverage} = \frac{|P \cap U|}{|U|},$$

We also measure the cosine similarity between the activation profiles of $P$ and $U$, where each profile is obtained by averaging the activations of all units in the basket over the probing set.

Over 50 sampled pairs, we obtain:

$$\text{parent\_coverage} : \mu = 0.94, \sigma = 0.07; \quad \text{union\_coverage} : \mu = 0.69, \sigma = 0.19;$$

$$\text{activation cosine} : \mu = 0.995, \sigma = 0.006.$$

Thus, almost all parent neurons (about 94% on average) are contained in the union of their children, and the resulting neuron sets exhibit very similar activation behaviour (cosine $\approx$ 0.995). At the same time, the lower union_coverage (about 69%) indicates that child baskets contain additional units beyond the parent, reflecting finer-grained or partially distinct concepts that are decomposed at lower levels of the hierarchy. Taken together, these results support the view that higher-level baskets can be faithfully interpreted as unions of their child baskets, while child baskets provide a more detailed factorization of the parent concept, consistent with the hierarchical, set-theoretic interpretation of Neurobasket.

## E BASKET FILTERING ANALYSIS

### E.1 FILTERING STATISTICS FOR LEVEL-1 BASKETS

To clarify the effect of our filtering step, we report the statistics for all *valid* level-1 baskets in ResNet-50 layer4 (before filtering). For each metric (cohesion, purity, simplified Davies–Bouldin

index $R_i$), we remove baskets whose scores lie above $\mu + 2\sigma$, treating them as extreme outliers that are likely to harm human interpretation rather than contribute meaningful concepts.

| Level | #valid baskets | cohesion fail | purity fail | $R_i$ fail |
|---|---|---|---|---|
| 1 | 207,902 | 543 (0.26%) | 1,271 (0.61%) | 6,654 (3.20%) |
| Pass all criteria | | | | 199,716 (96.06%) |

Table A.3: **Level-1 filtering statistics.** Only a small fraction of baskets are flagged as outliers by each metric, and more than 96% of baskets simultaneously satisfy all three criteria.

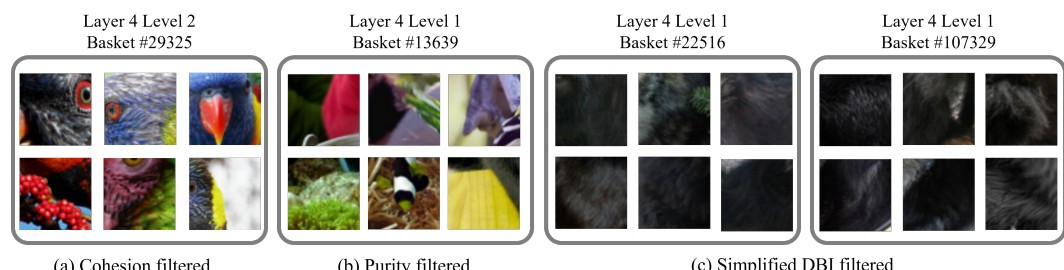

(a) Cohesion filtered     (b) Purity filtered     (c) Simplified DBI filtered

Figure A.9: **Examples of baskets removed by our filtering step.** (Left) *Low cohesion / purity.* The first two baskets contain several on-topic patches (e.g., parrot heads or clustered objects) but also many clearly off-topic regions such as unrelated berries, clothes, or background textures, indicating that the cluster does not represent a single coherent concept. (Right) *High redundancy (high $R_i$).* The last two baskets consist almost exclusively of very similar black–fur patches; although each basket is internally consistent, they are nearly indistinguishable from each other and thus considered redundant at this hierarchy level.

**Results.** As Table A.3 shows, our filtering is highly conservative: only 0.26% of level-1 baskets are removed due to low cohesion, 0.61% due to low purity, and 3.20% due to a high simplified $R_i$ score. Overall, 199,716 out of 207,902 baskets (96.06%) pass *all* criteria and are retained for subsequent experiments. In practice, the filtered baskets mostly correspond to visually noisy groups (mixed, unrelated concepts within a single basket) or near-duplicate clusters that are not well separated from their neighbors. Thus, the filtering step removes a small number of clearly uninterpretable or redundant outliers while preserving the vast majority of meaningful neuron groups used throughout our analyze.

### E.2 Qualitative Examples of filtered baskets

The examples in Fig. A.9 illustrate the types of baskets removed by our filtering procedure. Baskets with low cohesion or purity mix multiple unrelated visual patterns within a single group, which makes them hard to interpret as a single concept. Conversely, baskets with high $R_i$ capture concepts that are already represented by another nearby basket (e.g., duplicate "black fur" groups), so we discard them to avoid redundant explanations while keeping the vast majority of well-behaved, interpretable baskets.

## F LLM-based Abstraction Scoring Across Layers

To quantitatively support the claim that deeper layers encode more abstract concepts (Fig.6 and Fig. A.6–A.8), we additionally measure an abstraction score using an LLM-based evaluator. For each model and layer, we randomly sample 100 baskets and ask a GPT API to rate how abstract the underlying concept is, on an ordinal scale where larger values correspond to more abstract / higher-level descriptions. We perform this evaluation in two ways: (i) directly from the cluster images, and (ii) from the corresponding representative captions.

**Image-based abstraction scores.** For each basket, we randomly sample 30 member images and present them to the LLM with instructions to judge whether they depict low-level visual patterns

(e.g., edges, colors, textures) or higher-level semantic concepts (e.g., parts, objects, scenes), returning a single abstraction score per basket. Table A.4 reports mean scores over $N{=}100$ baskets per depth.

Table A.4: **LLM abstraction scores from cluster images.** For all three architectures, we mea

| Model | block 1 | block 2 | block 3 | block 4 |
|---|---|---|---|---|
| ResNet-50 | 2.53 | 2.91 | 2.92 | 3.08 |
| ResNet-18 | 2.54 | 2.96 | 2.98 | 3.01 |
| ViT-B/16 | 2.70 | 3.04 | 3.08 | 3.05 |

Across all architectures, abstraction scores increase from early to later blocks, indicating that baskets discovered from deeper layers correspond to more abstract concepts than those from shallower layers.

**Caption-based abstraction scores.** We repeat the same procedure using only the representative caption of each basket (instead of raw images), again sampling 100 baskets per depth. Table A.5 shows the mean abstraction scores.

Table A.5: **LLM abstraction scores from representative captions.** Deeper layers yield more abstract textual descriptions of baskets.

| Model | block 1 | block 2 | block 3 | block 4 |
|---|---|---|---|---|
| ResNet-50 | 2.74 | 3.04 | 3.28 | 3.38 |
| ResNet-18 | 2.32 | 2.63 | 2.82 | 2.81 |
| ViT-B/16 | 2.80 | 3.21 | 3.39 | 3.32 |

Both image-based and caption-based evaluations exhibit a consistent upward trend in abstraction score with depth, for ResNet-50, ResNet-18, and ViT-B/16. These results provide quantitative evidence that Neurobasket's baskets capture hierarchically evolving concepts, complementing the qualitative visualizations in Fig. A.6–A.8.

**Example of prompt for image abstraction score**

```
You are an expert in visual representation analysis.

You will see multiple images that all belong to a single cluster produced
↪  by some neural network layer or block.
All images in this cluster share some common visual feature.

Your job is to:
1. Look at all the images together.
2. Infer the main shared visual feature that best explains why these
↪  images belong to the same cluster.
3. Judge how abstract that shared feature is.
4. Summarize the concept with a short label and explanation.

Abstraction scale:
1 = very low-level visual properties
    (e.g., simple colors, edges, orientations, local contrast, simple
    ↪  textures, primitive shapes)
3 = mid-level visual patterns
    (e.g., parts of objects, local patterns, materials, characteristic
    ↪  textures like fur, grass, bricks,
     common arrangements of parts without a full object or scene concept)
5 = high-level semantic concepts
    (e.g., specific objects, object categories, scenes, activities,
    ↪  interactions, relationships between entities)

Guidelines:
```

```
- Focus on what is consistently shared across most images in the cluster.
- If the only consistent feature is color, brightness, or generic
↪  texture,
  treat it as low-level (closer to 1).
- If the consistent feature is about materials, parts of objects, or
↪  characteristic local patterns
  (e.g., "fur patches", "wheel rims", "brick walls"), treat it as
  ↪  mid-level (around 3).
- If the consistent feature is about whole objects, object categories,
↪  scenes, activities, or interactions
  (e.g., "people skiing on snowy mountains", "dog faces", "cars on a
  ↪  road"),
  treat it as high-level (closer to 5).
- Ignore accidental similarities that appear in only a few images.
- Base your judgment purely on the visual content of the images you see.
- Do NOT try to infer which layer or block produced the cluster; only
↪  consider the visual feature itself.

Return a single JSON object only, with this exact structure:

{
  "abstraction_score": 1,
  "label": "...",
  "reason": "..."
}

Where:
- "abstraction_score" is an integer from 1 to 5.
- "label" is a concise 2{5 word summary of the shared concept.
- "reason" is a 13 sentence explanation of why you chose that score,
↪  based only on the images.

Do not add any extra keys.
Do not include any explanatory text outside of the JSON.
Only output valid JSON.
```

