# OpenReview forum: "NeuroBasket: Interpreting Neuron Responses with Semantic Baskets"
_ICLR.cc/2026/Conference — Submitted to ICLR 2026_

### Official Review · Reviewer_fGnd · 2025-10-21

**Soundness:** 1
**Presentation:** 4
**Contribution:** 2
**Rating:** 2
**Confidence:** 4

**Summary:**

The paper introduces Neurobasket, a new explainable AI (XAI) method for concept extraction in deep neural networks. In Neurobasket, activations from a specific layer are collected using images from a probing set and grouped through hierarchical clustering. To make sure the resulting clusters are meaningful, they are filtered based on several properties, including caption-embedding purity (L. 163).

These clusters are then matched with human-understandable concepts using vision-language and large language models. Finally, the clusters are discretized, meaning each one is linked to a specific set of neurons rather than a continuous direction in feature space. This is done by identifying which neurons are active for images that belong to a given Neurobasket.

The experiments show that Neurobasket identifies coherent neuron groups. They also demonstrate that unions and differences between Neurobaskets correspond well with human intuition.

**Strengths:**

- The paper is well written and easy to follow.
- The experiments cover multiple network backbones and datasets.
- The paper addresses an important problem and aligns with the current direction of the field, emphasizing the need to move beyond single-neuron explanations.
- The hierarchical approach is interesting, and it is appealing that unions and differences of concepts behave as intuitively expected.
- The appendix is detailed, providing essential experimental settings, and the authors also release the code, which supports reproducibility.

**Weaknesses:**

- **W1 Related Work.** In my opinion, the paper overlooks several important related works. Line 105 only mentions *FALCON* and *NeurFlow* in the context of MNEs, but omits studies that analyze feature space *directions* such as *CRAFT* (Fel et al., 2023) and *Sparse Autoencoders* (Cunningham et al., 2023). I also do not see a strong justification for why directions should fundamentally differ from sets of neurons.
  Moreover, the idea of hierarchical representations has already been explored, for example in *Matryoshka SAEs* (Bussmann et al., 2025). The paper should either make a clear argument for why using neuron sets instead of directions is important, which is currently missing, or more explicitly position its contributions within the context of existing related work.

- **W2 Insufficient quantitative evaluation.** For many of the experiments, the paper only presents a handful of qualitative examples, which is not sufficient to properly support claims such as that Neurobasket "supports set-theoretic reasoning" (L. 86), that it offers "a more faithful and systematic view of neural representations" (L. 485), or that "explanations [...] are structurally organized and relationally grounded, offering deeper insights into how complex concepts are encoded" (L. 92). Also, following W1, relevant baselines for concept extraction through directions should be included for comparison.
Furthermore, the provided quantitative evaluations do not effectively assess the interpretability or utility of the proposed method. The analysis in Sec. 4 primarily checks whether the activations of the selected neurons are similar, which is desirable but not directly related to interpretability. In addition, the pruning evaluation in Sec. 5.3 is not compared to established methods for concept extraction, making it difficult to place the results in context.

- **W3 Hyperparameters are not thoroughly evaluated.** The method involves several hyperparameters, such as alpha and beta, but it is not clear how these affect the interpretability or overall results of the method. While the appendix (B.3) provides a brief evaluation of alpha, this only covers the proposed activation consistency metric and does not assess interpretability directly.

- **W4 Filtering could remove important concepts.** The proposed method includes a filtering step that removes "uninterpretable" clusters (L.162). While this helps produce cleaner results, it raises the question of how faithful or complete the resulting explanations truly are.


**Minor Weaknesses**
- There are a few typos, e.g., L. 452, 425, and 459.
- I only see experiments supporting the *union* and *difference* operations (Sec. 5.1 and 5.2); however, in L. 87, *intersection* is also mentioned as a contribution.
- In Table 1, the standard deviation for *FALCON* is missing.
- The capitalization of the method’s name in the title is inconsistent with its usage in the main text (NeuroBasket vs. Neurobasket).

**Questions:**

- How does the method fit into the broader scope of concept extraction, including approaches based on direction analysis? If there is a clear unique selling point for neuron sets over directions, please provide supporting arguments or empirical evidence.
- How can the interpretability and utility of the proposed method be evaluated, and how would it compare to existing approaches for concept extraction?
- How sensitive is the method to different hyperparameter settings?
- How many clusters are removed through the filtering step, and how does this affect the completeness of the resulting explanations?

While the paper is a nice read and presents interesting initial qualitative results, I feel it is still too far from the acceptance threshold, primarily due to the lack of quantitative metrics evaluating the interpretability and utility of the proposed method and comparing it to baselines using directions. Addressing these issues would probably require substantial additional analyses, which I consider beyond the scope of a standard review cycle.

I thank the authors for their effort and look forward to reading the rebuttal.

---

> ### Author Response · Authors · 2025-11-22
> **Rebuttal for Reviewer fGnd**
>
> We really appreciate your thoughtful comments and suggestions for our paper.
>
> A1. Related Work
>
> Thank you for pointing out these missing references and for raising the question about neuron sets vs. directions. We expanded the related-work section to explicitly discuss CAV/CRAFT-style concept directions and recent SAE-based approaches, including Matryoshka SAEs. Concept-direction methods such as CAV/CRAFT typically operate in the feature space of a single layer, learning dense vectors where most coordinates have non-zero weights. This is very effective for detecting whether a concept is present along a particular direction, but it makes it difficult to (i) isolate how that concept contributes to the model’s final decision and (ii) analyze how it interacts with other concepts or groups of neurons, because each direction blends many units into a continuous combination. In contrast, our work explicitly groups concrete neuron indices into baskets across layers and then studies how the model represents concepts as combinations of these baskets, as well as how adding/removing baskets (union/difference, ablation) changes prediction behaviour.
>
> SAE-based works, including Matryoshka SAEs, are closer in spirit but still have a different focus and objective. SAEs learn a new sparse latent basis that “unwraps” the representation into interpretable directions, often optimized for reconstruction or sparsity. Matryoshka SAEs further propose a hierarchical structure on these latent codes, but the hierarchy lives in the learned representation space, not in the space of original neurons grouped by their behaviour across images. In contrast, we do not learn a new representation; we directly group existing neurons into baskets grounded in activation statistics over an image hierarchy and then analyze how these baskets, and their unions/differences, affect the network’s behaviour and semantics. We clarified this in the revised related-work section.
>
> A2. Insufficient quantitative evaluation.
>
> Thank you for the insightful comments. We agree that some of our statements were too strong and will soften the wording around “set-theoretic reasoning” and “more faithful/systematic view”. In the revision, we explicitly state that NeuroBasket supports set-operation–based analysis as a practical analysis tool, rather than claiming a fully formal set-theoretic framework.
>
> Regarding quantitative evidence, our goal is to show that NeuroBasket produces (i) stable neuron groups, (ii) with causal influence on predictions, and (iii) semantically coherent summaries:
>
> The consistency metrics in Sec. 4 measure whether neurons within a basket behave coherently across images and architectures; without this stability, any explanation would be unreliable.
> The pruning experiments in Sec. 5.3 Then connect these groups to behaviour by showing that ablating a basket induces structured, non-random changes in class probabilities.
>
> To more directly address the faithfulness of the textual summaries, we add two new embedding-based evaluations:
>
> Sentence-BERT similarity: For 10,000 sampled baskets, we compute cosine similarity between each basket’s representative summary and the image captions used to form that summary, versus captions chosen at random (same count). Our pairs achieve 0.5623 ± 0.1121, while random pairs achieve 0.2238 ± 0.0645.
>
> CLIP image–text similarity: For 10,000 sampled layer-4 baskets, we compare CLIP similarity between basket images and their own representative summary versus random summaries. We obtain 0.2650 ± 0.025 vs. 0.2020 ± 0.014 for random.
>
> These results, together with the activation consistency and pruning analyses, provide quantitative support that our baskets are stable, causally meaningful units whose summaries are substantially more aligned with the underlying images than random text.
>
> A3. Hyperparameters are not thoroughly evaluated.
>
> We appreciate the concern about the role of the hyperparameters α and β. In our framework, α and β are only used in the basket construction stage to decide which neurons are included or excluded in each basket based on activation statistics (i.e., how strongly and how selectively a unit responds). Importantly, our claims about interpretability do not rely on tuning α or β or on any VLM/LLM-specific settings. The VLM captions and LLM summaries are always applied to a fixed set of baskets, and we added to quantitatively assess their semantic faithfulness using independent alignment metrics (i.e., Sentence-BERT similarity, CLIP similarity).

---

> ### Author Response · Authors · 2025-11-22
> **Rebuttal for Reviewer fGnd**
>
> A4. Filtering could remove important concepts.
>
> We agree that overly aggressive filtering could in principle discard meaningful concepts. In our framework, however, the filtering step is intentionally very conservative and mainly removes clusters that are clearly unhelpful for interpretation.
>
> 1) Only ~4% of clusters are filtered out.
>
> For example, at level-1 on ResNet-50 layer4 we have 207,902 valid clusters; 96.1% of them pass all three criteria (cohesion, purity, DB-index). The failure rates per metric are each well below 4%. Thus, the default behavior is to keep clusters, and filtering is applied only to a small tail of outliers.
>
> 2) What is actually removed?
>
> Manual inspection shows two typical patterns:
>
> (i) Noisy/mixed clusters (low cohesion or purity): e.g., clusters that simultaneously contain “blue bird eye,” red berries, and toys, or lobster images mixed with human legs and clothes. Another example contains stripes, human necks, fish, and reptile skin all in one group. These clusters have no single coherent concept even for human annotators.
>
> (ii) Redundant duplicates (high DB-index): pairs of clusters that both contain almost identical “finger” patches or “black fur” patches, with very similar cohesion/purity and almost zero between-cluster separation. Dropping one of them does not reduce concept coverage; it only avoids double-counting.
>
> 3) Impact on explanations.
>
> Even if we keep the filtered clusters, the main quantitative trends remain unchanged. This suggests that our explanations are driven by the large majority of stable, coherent clusters, and not by the small, noisy/redundant tail that is filtered out.
> We clarified these points in the revised appendix with statistics and a few visual examples of filtered clusters in Appendix E.

---

> > ### Comment · Reviewer_fGnd · 2025-11-27
> >
> > I thank the authors for the detailed feedback and for updating the manuscript. Since some of my concerns have been addressed, I increased my score. However, after reading the other reviews, I am still not completely convinced to recommend the paper for acceptance. My main concerns include:
> >
> > - W1/W2: I understand the argument that existing direction-based approaches only work in a single layer. However, I am still missing a clear demonstration of the advantages compared to existing work. The qualitative examples showing that set-theoretic operations are doable are nice, but I am not sure if they increase my understanding of DNNs in any way or help with specific downstream applications. The included analysis on the faithfulness is highly appreciated.
> >
> > - W3: For the hyperparameters, I do not fully get the argument. Even if $\alpha$ and $\beta$ are only used in the basket construction stage to decide which neurons are included or excluded in each basket, the specific choices would have a strong effect on the obtained explanations, and how to select them or which impact they have remains unclear.
> >
> > - W4: The filtering argument is good, and that concern can be removed from my weaknesses.

---

> > > ### Author Response · Authors · 2025-12-03
> > >
> > > Thank you again for carefully reading our work and participating discussion. Below, we address your remaining concerns.
> > >
> > > ### On W1/W2: advantages over direction-based work and usefulness of set-operations
> > >
> > > We agree that simply demonstrating union/difference operations is not enough; what matters is what they reveal beyond existing approaches.
> > >
> > > Direction-based methods (e.g., CAV/CRAFT, SAEs) are effective tool for defining and detecting concept directions in a given feature space: one can clearly say “this direction corresponds to concept c.” However, because each direction is a dense combination of many units, it is often difficult to (i) track how this concept actually affects the decision, (ii) understand how multiple concepts are combined, mixed, or even distributed as the network moves from intermediate features to final decisions.
> > >
> > > Single-neuron explanation (SNE) methods sit at the opposite extreme: they make the role of individual units very clear (for example, “this neuron responds to dog faces”), but they make it hard to see how many such units should be interpreted together, or how collections of neurons jointly support a concept and its causal effect on predictions.
> > >
> > > Neurobasket is designed specifically to bridge this gap. Instead of starting from directions or from hand-picked single neurons, we first define concept-level baskets in image space via hierarchical clustering, and then select neuron sets that respond consistently to each basket based purely on activation statistics. This yields a basket-first library of explicit neuron indices that (i) can be intervened on directly via ablation and (ii) are organized hierarchically across abstraction levels.
> > >
> > > On top of this basis, we perform set-operation–based analysis:
> > >
> > > - Union of child baskets: We quantitatively show that the explicit union of child baskets closely approximates the parent basket. Across sampled parent–child groups, the union covers about 94% of the parent units on average, and the cosine similarity between the parent’s activation profile and the union’s profile is about 0.995. This suggests that higher-level concepts behave as compositional combinations of lower-level neuron sets, giving a concrete handle on how concepts are built up.
> > >
> > > - Difference between overlapping baskets: by ablating only the residual units in A \ B, we can probe fine-grained, discriminative cues that drive specific changes in class probabilities, separating shared structure from basket-specific semantics that are hard to see from clustering alone.
> > >
> > > While our framework is primarily an analysis tool rather than a task-specific detector, these operations let us move beyond “which concept is present” (direction-based) or “what does this neuron do” (SNE), toward “which concrete neuron sets jointly implement a concept, how are they composed, and how does intervening on them change the model’s behaviour.”
> > >
> > > ### On W3: sensitivity to neuron-selection hyperparameters
> > >
> > > We appreciate the emphasis on hyperparameter sensitivity. In the revised manuscript, we add ablations over both the activation threshold alpha and the consistency threshold beta, and measure their effect on:
> > >
> > > - mean number of neurons per basket
> > > - activation consistency (mean off-diagonal cosine similarity)
> > > - mean change in the true-class logit per image under basket ablation
> > >
> > > For ResNet-50 layer 4, varying alpha in {2, 3, 5, 7} (with beta = 0.75) gives the following trends:
> > >
> > > - As alpha increases, baskets become smaller and more internally consistent (activation consistency increases from 0.586 to 0.707).
> > > - At the same time, the mean per-image true-class logit drop under ablation decreases smoothly from 0.546 to 0.121.
> > >
> > > Thus, increasing alpha trades off a stronger causal impact against higher within-basket consistency. Our choice alpha = 3 lies in the middle of this continuum: neuron sets are substantially more consistent than at alpha = 2, while still inducing relatively large and structured changes in class logits when ablated.
> > >
> > > Similarly, for beta in {0.5, 0.7, 0.75, 0.8, 0.9, 1.0} (with alpha = 3), we observe:
> > >
> > > - Increasing beta makes baskets smaller, and activation consistency increases (from 0.588 to 0.675).
> > > - The mean true-class logit drop under ablation decreases from 0.558 to 0.126 as beta increases.
> > >
> > > Overall, these ablations show that:
> > >
> > > - The behavior of Neurobasket with respect to $\alpha$ and $\beta$ is smooth and not sensitive.
> > > - There is a clear, controllable trade-off between intra-basket homogeneity and the magnitude of causal impact.
> > > - The baseline setting ($\alpha = 3$,  $\beta = 0.75$) is a balanced choice that preserves both high consistency and substantial influence on predictions.
> > >
> > > We hope these additional analyses clarify both what Neurobasket adds beyond direction-based approaches and how stable the neuron-selection mechanism is under hyperparameter variation.

---

### Official Review · Reviewer_B6Vw · 2025-10-30

**Soundness:** 2
**Presentation:** 2
**Contribution:** 2
**Rating:** 2
**Confidence:** 5

**Summary:**

The paper proposes NeuroBasket, an interpretability framework that groups neurons into semantically coherent sets called "baskets" through hierarchical clustering and natural language grounding. It aims to move beyond single-neuron interpretability and non-hierarchical grouping methods.

**Strengths:**

- The paper articulates a real limitation in interpretability: single-neuron explanations fail when representations are highly distributed.
- Constructing a hierarchy over neuron groups and enabling set-theoretic operations (union/difference) is a useful conceptual advance.

**Weaknesses:**

1. The novelty of this paper is limited. Hierarchical grouping of neurons is not a new problem and has already been studied in previous works, such as (Wang et al., 2022), NeurFlow (Cao et al., 2025). The authors should clearly justify the key differences in their approach compared to other hierarchical grouping approaches. What are the new findings and justify the novelty.
2. For each basket, consistently active neurons are identified using β = 0.75. β (the hyperparameter for neuron set selection) is a key hyperparameter, but the choice of this value seems ad hoc and lacks theoretical or experimental justification. The authors should justify their choice of β.
3. Most experimental results are limited to comparisons with FALCON and a Random-index baseline. The authors should also include evaluations against other Multi-Neuron Explanation (MNE) methods, as well as representative Single Neuron Explanation (SNE) approaches (e.g., Clip-dissect, WWW). Such comparisons are essential to (1) justify the claimed improvements over traditional SNE methods and (2) demonstrate improvement compared to previous MNE frameworks.
4. The approach assumes that VLM captions and LLM summaries correctly reflect the underlying neuron or feature cluster semantics. However, VLM captions often contain hallucinations or bias from training data. So, the “human-understandable meaning” may not faithfully represent what the cluster actually encodes. Failure in this step undermines the entire interpretability pipeline.

- Wang, Andong, Wei-Ning Lee, and Xiaojuan Qi. "Hint: Hierarchical neuron concept explainer." Proceedings of the IEEE/CVF Conference on Computer Vision and Pattern Recognition. 2022.
- Cao, Tue Minh, et al. "NeurFlow: Interpreting Neural Networks through Neuron Groups and Functional Interactions." The Thirteenth International Conference on Learning Representations. 2025

**Questions:**

Same as weaknesses.

---

> ### Author Response · Authors · 2025-11-22
> **Rebuttal for Reviewer B6Vw**
>
> We really appreciate your thoughtful comments and suggestions for our paper.
>
> A1. About limited novelty.
>
> We thank the reviewer for raising this question. Our contribution is not just “another hierarchical grouping,” but a basket–first framework for analyzing model behavior.
>
> First, instead of starting from pre-defined concepts or individual neurons, we construct a stable library of neuron baskets by clustering images and then selecting units that respond consistently to each image group based purely on activation statistics. The hierarchy is defined in image space (via FINCH), and baskets are attached to each node as its most stable responders. This basket–first, data-driven construction is decoupled from any external ontology or text supervision, so the resulting neuron groups form a reusable internal basis for subsequent analysis rather than being tied to a specific concept set.
>
> Second, we introduce a novel set-operation–based analysis from a basket-level view of model behavior. In set-operation–based analysis, we use the union and difference of baskets as an interpretive lens for compositional changes in the network. We quantitatively verify that the explicit union of child baskets closely approximates the parent basket in both index coverage and activation similarity. This shows that our hierarchy supports meaningful compositional reasoning about the model’s behavior grounded in activation statistics.
>
> A3. Comparison with other baselines.
>
> Our intention is not to position NeuroBasket as a replacement for single-neuron explanation (SNE) methods such as CLIP-dissect or WWW, but as a higher-level framework that can be built on top of them. SNE approaches start from pre-defined concept vocabularies (e.g., WordNet) and map individual neurons to human labels. By contrast, our work assumes the model’s important behavior is distributed (as illustrated in Fig. 1) and asks a different question:
>
> 1. Which neurons should be grouped and interpreted together?
>
> 2. How stable are these groups?
>
> 3. How do such groups affect the model’s behaviour?
>
> In other words, SNE is a tool for understanding single units, while NeuroBasket provides a basket-first, hierarchical, and operation-level view of neuron groups. This also underlies our novelty claim. Rather than starting from concept labels or hand-picked neurons, we first construct a stable library of neuron baskets by clustering images and selecting units that respond consistently to each cluster, and then attach these baskets to a FINCH-based image hierarchy. On top of this library, we introduce a set-operation–based analysis of model behaviour, where union and difference of baskets are used as an interpretive lens and are quantitatively linked to activation similarity and causal effects on predictions. Existing SNE and MNE methods do not provide this kind of basket-level stability analysis and compositional reasoning over neuron groups.
>
> Because of this difference in target, we did not consider CLIP-dissect / WWW / HINT as direct baselines, but rather as complementary components. Any SNE method could be plugged into our framework to annotate individual neurons to extend concepts at the unit level. In the revised manuscript, we make this positioning clear in the related-work and discussion sections, and clarify that our FALCON and Random-index baselines are used to evaluate the group-level selection and causal impact, while SNE methods can be used as additional tools that can expand NeuroBasket with detailed concepts at the unit level.
>
> A4. Concerns about VLM / LLM Faithfulness.
>
> The core pipeline (i.e., constructing baskets) is defined entirely in activation/probability space and runs independently of any VLM/LLM. The textual layer is added only afterwards as a human-facing interface; even if some captions are noisy or biased, this does not affect how baskets are selected or how the quantitative results are obtained.
>
> That said, we agree it is important to check that the generated text is at least meaningfully related to the associated clusters. In the revision, we therefore add two embedding-based evaluations:
>
> (i) Sentence-BERT similarity between each basket’s summary and the image captions used to form that summary versus random captions, where our pairs achieve mean cosine similarity 0.56 ± 0.11 compared to 0.22 ± 0.06 for random.
>
> (ii) CLIP image–text similarity between basket images and their representative summaries versus random captions, where our pairs achieve 0.27 ± 0.03 versus 0.20 ± 0.01 for random.
>
> These gaps indicate that, while we do not treat the summaries as perfect semantic ground truth, they are substantially more aligned with the underlying image clusters than chance, and thus provide a reasonably faithful, though imperfect, “human-understandable” view of the baskets. Full details are included in the appendix C.

---

### Official Review · Reviewer_7sy9 · 2025-11-01

**Soundness:** 2
**Presentation:** 2
**Contribution:** 2
**Rating:** 4
**Confidence:** 3

**Summary:**

The authors proposed Neurobasket, a hierarchical framework for interpreting neural networks by grouping neurons into semantically coherent sets called *baskets*.
It first applies hierarchical clustering using FINCH to activation features and grounds each cluster in natural language using captions from a vision–language model and LLM summarization.
Neurons that consistently activate across a cluster’s images are then assigned to form stable basket representations.
These neuron groups enable set-theoretic-like analyses (union, difference) to explore compositional structure in model representations. Experiments on ResNet and ViT models show that the proposed produces somewhat coherent, semantically aligned neuron sets that generalize across architectures and datasets.

**Strengths:**

- **S1. Practical and Coherent Integration of Existing Multimodal Tools for "Basket"**

The definition of a basket reflects a fair and coherent integration of existing techniques rather than introducing new algorithms.
The framework combines hierarchical clustering with vision–language captioning and LLM summarization in a straightforward and practical manner.
This design effectively leverages multimodal and language models to associate neural activations with interpretable semantic descriptions without adding unnecessary complexity.
While not conceptually novel, the integration is technically sound and intuitively organized, making the framework easy to implement and extend to various architectures or datasets.

- **S2. Fair and Feasible Experimental Design Demonstrating Applicability**

The experimental setup is reasonably designed to demonstrate the applicability of the proposed framework across different architectures and datasets.
In the absence of well-established baselines for multi-neuron interpretability, the inclusion of random-index and random-basket comparisons provides a reasonable reference for evaluating activation consistency and causal effects.
This pragmatic approach ensures that the reported improvements of the proposed method are interpreted in context rather than overstated.
Additionally, testing on both convolutional and ViT models helps illustrate the generality of the method, reflecting a fair and balanced effort to assess its robustness under diverse conditions.

**Weaknesses:**

- **W1. Limited Quantitative and Stable Evaluation of "Baskets"**

The concept of a "basket" in the proposed method--linking clustered neural activations with textual descriptions--offers an intuitive bridge between network activity and human-understandable semantics.
However, because the semantic component is expressed only through LLM-generated text, it cannot be quantitatively compared or verified, leaving the evaluation of semantic coherence largely qualitative.

To enhance rigor and reproducibility, the authors could include quantitative analyses using semantic embeddings derived from basket descriptions.
Embedding the captions or summaries into a multimodal space (e.g., Sentence-BERT) would allow measuring inter-basket similarity, quantifying hierarchical abstraction, and testing whether neuron-set operations correspond to meaningful semantic changes.
Additionally, as the current grounding pipeline depends on captioning and LLM summarization, the resulting descriptions may vary with prompting or model choice. Embedding-based representations or text–image alignment checks could make the semantics more stable and verifiable.

- **W2. Conceptual and Empirical Ambiguity in Set-Theoretic Reasoning**

The paper highlights set-theoretic reasoning--particularly the union and difference of neuron sets--as a central interpretive feature of the proposed method. However, the implementation~(Fig. 4) appears more illustrative than strictly set-theoretic.
Especially, the "union" between child baskets does not functionally generate the parent basket through the explicit combination of neuron sets; rather, the parent representation is obtained from an independently formed higher-level cluster.
Consequently, the demonstrated relationship reflects qualitative alignment within the hierarchy rather than an actual compositional process where $S_p \approx S_a ∪ S_b$.

To substantiate this claim, the authors could quantitatively verify whether the union of child neuron sets reproduces or approximates the parent basket’s activation profile or semantic embedding.
Demonstrating this correspondence across activation and semantic spaces would clarify whether hierarchical abstraction truly arises from compositional structure rather than coincidental clustering.
Such validation would make the concept of set-theoretic reasoning both theoretically precise and empirically grounded.

- **W3. Ambiguity and Limited Validation in Qualitative Visualization**

Fig. 6 aims to illustrate how the proposed approach captures progressively abstract concepts across model depth, yet the notion of level remains only partly clarified.
Fig. 2 indicates that level refers to the hierarchical depth within the clustering process, but in later figures the term is used alongside network layers (e.g., Layer 2-Level 3), blurring whether it denotes clustering granularity, architectural depth, or both.
This conflation makes it difficult to interpret how hierarchical organization interacts with model structure to yield higher-level semantics.

Although the authors state that each image group corresponds to a specific basket, the alignment between the visual exemplars and their textual descriptions is not empirically validated. It remains uncertain whether the images faithfully capture the semantic content expressed in the accompanying summaries.
To substantiate the claim that deeper layers yield more abstract and coherent representations, the authors could complement Fig. 6 with quantitative or human-based evaluations of image–text correspondence (e.g., CLIP similarity or participant agreement).
Such analyses would strengthen the evidence that the proposed method captures genuinely human-interpretable hierarchical abstractions rather than relying on visual illustration alone.

**Questions:**

Most of my main concerns or questions have been outlined in the Weaknesses section.
I listed some additional questions here.

- Q1. If the textual grounding was primarily intended for interpretability, how might the authors ensure reproducibility in future implementations given LLM non-determinism?

- Q2. How sensitive are the results to hyperparameters such as the activation threshold ($\alpha, \beta$) used in neuron selection?

---

> ### Author Response · Authors · 2025-11-22
> **Rebuttal for Reviewer 7sy9**
>
> We really appreciate your thoughtful comments and suggestions for our paper.
>
> A1. Limited Quantitative and Stable Evaluation of "Baskets"
>
>
> We agree that it is important to clarify what is actually “grounded” in our framework. Our main contribution is a stable construction of neuron groups (baskets) based on activation statistics and clustering; all key quantitative results (consistency, causal pruning, cross-architecture/dataset generalization) are defined purely in activation/probability space and do not depend on any LLM. The LLM-generated summaries are intended only as additional interpretable information to already-fixed baskets, not as ground-truth semantic labels, so the specific LLM choice or prompt does not affect the concept representation of neuron groups.
>
> That said, we agree that it is useful to quantitatively check that these summaries are at least coherent with the associated images. In the revised version, we add two embedding-based evaluations: (1) Sentence-BERT similarity between each basket’s summary and the image captions used to form that summary vs. random captions, where our pairs achieve mean cosine similarity 0.56 ± 0.11 compared to 0.22 ± 0.06 for random; and (2) CLIP image–text similarity between basket images and their summaries vs. random captions, where our pairs achieve 0.27 ± 0.03 vs. 0.20 ± 0.01 for random. In both cases, the summaries show substantially higher similarity than random baselines, supporting that images and summarized texts are meaningfully aligned with each other. Full details and experimental settings are provided in the appendix C.1 and C.2.
>
> A2. Conceptual and Empirical Ambiguity in Set-Theoretic Reasoning
>
>
> Our hierarchy is built with FINCH clustering over images, so each parent cluster is formed from a superset of the images belonging to its child clusters, and baskets are then selected based on consistent activations within each cluster. Our use of “set-theoretic” language is therefore meant as an interpretive lens motivated by set unions and differences, rather than a claim of mathematically exact set algebra over neuron indices. We revised expressions in the paper to make this intent clear and avoid over-claiming.
> In addition, we provide a quantitative evaluation comparing each parent basket to the explicit union of its child neuron sets. Across three representative parent–child groups, the coverage of the parent by the union of its children averages about 0.96, and the cosine similarity between the activation profiles of the parent and the child-union averages about 0.98. These results indicate that, in activation space, parent baskets are very well approximated by the union of their children. Full details of this analysis are included in the appendix D.
>
> A3. W3. Ambiguity and Limited Validation in Qualitative Visualization
>
>
> In the revision, we now clearly distinguish the network layer index 𝐿 from hierarchical depth 𝐻 of the clustering tree. We explicitly state this notation in the method section and clarify in the captions of Figures that “level refers to the hierarchical depth in the clustering tree.”
> To support the results of Fig. 6 quantitatively, we add two analyses. (i) CLIP image–text alignment. Using 10k sampled baskets from ResNet-50 layer 4, we compute CLIP similarity between each basket’s images and its own representative caption vs. random captions. The mean cosine similarity is 0.2650 ± 0.025 for true image–caption pairs, compared to 0.2020 ± 0.014 for random pairs, indicating that the displayed summaries are meaningfully aligned with their image groups. (ii) Abstraction score across depth. We use an LLM to rate the abstraction (higher = more abstract) of cluster image sets and representative captions. For ResNet-50, the scores increase with depth as follows (ResNet-18 and ViT show the same trend):
>
> Image abstraction for Layer block 1-4 : 2.53 / 2.91 / 2.92 / 3.08
>
> Caption abstraction for Layer block 1-4 : 2.74 / 3.04 / 3.28 / 3.38
>
> Taken together, the CLIP analysis provides empirical evidence that the image groups and their textual summaries form coherent pairs, while the GPT-based abstraction scores, which increase with depth, offer quantitative support that deeper layers tend to capture more abstract concepts, beyond the qualitative illustration in Fig. 6. Full details are included in the appendix F.

---

> ### Author Response · Authors · 2025-11-22
> **Rebuttal for Reviewer 7sy9**
>
> A4. Additional questions
>
> Q1. If the textual grounding was primarily intended for interpretability, how might the authors ensure reproducibility in future implementations given LLM non-determinism?
>
> A4-1. As we clarified in A1, our main contribution is a stable construction of neuron groups (baskets) based on activation statistics and clustering, which does not depend on any LLM. The LLM-generated summaries are used only as an additional interpretability layer for already-fixed baskets, not as ground-truth semantic labels, so the particular LLM or prompt does not affect the underlying concept representation.
> For reproducibility of the textual side, we fix the LLM version, the prompt template, and the summarization pipeline, and we already provide the corresponding code and prompts in the supplementary material. We can also release the generated text summaries upon request. This way, future implementations can either (i) reuse our exact summaries, or (ii) run the same pipeline with the model and prompts to obtain comparable textual descriptions.

---

> > ### Comment · Reviewer_7sy9 · 2025-11-25
> >
> > Thank you to the authors for the detailed and clarifying responses. They were very helpful.
> >
> > One remaining point I am still curious about concerns Q2, regarding the sensitivity of the hyperparameters used in neuron selection (such as the activation thresholds ($\alpha, \beta$).
> > This aspect was not addressed in the rebuttal.
> >
> > As other reviewers have also noted, an ablation study examining how these hyperparameters affect basket construction and downstream metrics (e.g., activation consistency, pruning impact, semantic coherence) would substantially strengthen the paper.
> > Incorporating such an analysis into the manuscript would help demonstrate the stability and robustness of the proposed neuron-selection mechanism.

---

> > > ### Author Response · Authors · 2025-12-03
> > >
> > > Thank you again for your follow-up and for the importance of understanding the sensitivity of the neuron-selection hyperparameters $\alpha$ and $\beta$. We agree that this is important and have added an explicit ablation study in the revised Appendix (Sec. B.3), focusing on how these parameters affect basket construction and downstream metrics.
> > >
> > > ### $\alpha$, $\beta$ ablation (Appendix B.3)
> > >
> > > We vary $\alpha$ (per-image activation threshold) and $\beta$ (fraction of images in a cluster that must activate a neuron) on ResNet-50 layer4 baskets, and for each setting, we measure:
> > >
> > > - Mean number of neurons per basket (size of neuron sets)
> > > - Activation consistency (mean off-diagonal cosine similarity across neuron activation profiles)
> > > - Causal impact (mean $\Delta y_{gt}$ per image under basket ablation, as in Sec. 5.3)
> > >
> > > Effect of $\alpha$ (with $\beta = 0.75$):
> > >
> > > | $\alpha$ | mean #neurons | consistency | mean $\Delta y_{gt}$ |
> > > |-----------|---------------|------------|------------------------------------|
> > > | 2.0       | 37.19         | 0.586      | 0.546                              |
> > > | 3.0       | 18.52         | 0.626      | 0.350                              |
> > > | 5.0       | 6.25          | 0.700      | 0.134                              |
> > > | 7.0       | 3.15          | 0.707      | 0.121                              |
> > >
> > > As α increases, baskets become smaller and more internally consistent, while their average causal impact decreases. Lower $\alpha$ retains more neurons per basket (stronger effect on the model but less homogeneous), whereas higher $\alpha$ yields very tight, high-consistency neuron sets that perturb predictions less. Our default $\alpha = 3$ lies in the middle of this trade-off, simultaneously achieving a clear consistency gain over $\alpha = 2$ while still inducing substantial probability drops.
> > >
> > > Effect of $\beta$ (with $\alpha = 3$):
> > >
> > > | $\beta$ | mean #neurons | consistency | mean $\Delta y_{gt}$ |
> > > |----------|---------------|------------|------------------------------------|
> > > | 0.50     | 31.49         | 0.588      | 0.558                              |
> > > | 0.70     | 20.03         | 0.620      | 0.405                              |
> > > | 0.75     | 18.52         | 0.626      | 0.350                              |
> > > | 0.80     | 17.25         | 0.633      | 0.273                              |
> > > | 0.90     | 11.35         | 0.667      | 0.192                              |
> > > | 1.00     | 10.40         | 0.675      | 0.126                              |
> > >
> > > We observe a similar pattern: increasing $\beta$ enforces that neurons fire on a larger fraction of cluster images, which raises consistency and sparsity but reduces the average pruning effect. The chosen $\beta = 0.75$ is where baskets are already clearly more consistent than at $\beta = 0.5$ or $0.7$, while still producing strong changes in class probabilities.
> > >
> > > Overall, these ablations show that:
> > >
> > > - The behavior of Neurobasket with respect to $\alpha$ and $\beta$ is smooth and not sensitive.
> > > - There is a clear, controllable trade-off between intra-basket homogeneity and the magnitude of causal impact.
> > > - The baseline setting ($\alpha = 3$,  $\beta = 0.75$) is a balanced choice that preserves both high consistency and substantial influence on predictions.
> > >
> > > Regarding semantic coherence, basket construction itself is purely activation-based; the semantic evaluations (Sentence-BERT and CLIP similarity) are reported for the default configuration and show that the resulting summaries are meaningfully aligned with their image clusters.
> > >
> > > We have added this ablation and its discussion to the revised appendix, and we hope it clarifies the stability and robustness of the proposed neuron-selection mechanism, directly addressing your question about hyperparameter sensitivity.

---

### Official Review · Reviewer_BvRj · 2025-11-02

**Soundness:** 3
**Presentation:** 3
**Contribution:** 2
**Rating:** 4
**Confidence:** 5

**Summary:**

The paper addresses the problem of neuron grouping in Multi-Neuron Explanation (MNE) using hierarchical clustering. The paper enables set-theoric reasoning and hierarchical organization, which is original, using the neuron basket.

**Strengths:**

The paper provide concrete criteria on neuron activation and selection for each basket / group which previous works don't. The authors also propose a "consistency" metric to quantitatively compare the grouping quality between the basket and previous works. Furthermore, the proposal of set-theoric reasoning is original and interesting.

**Weaknesses:**

This paper lacks comparison experiment about the quality of clustering beyond "consistency":

1, It would improve my score considereably if the author can provide further comparison between ablating / pruning neuron groups from [1] and [2] in the experiment of "causal pruning evaluation". Showing that this paper is more effective at selecting neurons that is causally important for the prediction of the model.

2, Beyond consistency, we may interest in other criterion such as "separatability", etc., between neuron groups. The consistency only shows  intra-level relation of one group, while separatability shows inter-level relation between multiple groups.

3, Could the author provide additional experiments to support the proposal of filtering metrics used in the main method (as discussed in Appendix B.1) on the quality of the neuron groups. Specifically, how do they affect the "consistency" and beyond, and how it is advantageous compared to previous works [1], [2]...

4, It is interesting to see the application of set-theoric reasoning in analyzing the effect of neuron groups on the model prediction, showing the usefulness of the proposal. For example, ablating higher level group (based on group hierarchy) affect more generalized concepts, while ablating set-difference would affect only a part of the concepts.

5, Minor comment: In the experiment measuring the "consitency", does the difference in the number of clusters for each method affect the outcome, for example, [1] uses 344 groups while the proposal method uses 10000 groups?

**Questions:**

Please refer to the Weaknesses section.

Citation:

[1]: Neha Kalibhat, Shweta Bhardwaj, C Bayan Bruss, Hamed Firooz, Maziar Sanjabi, and Soheil Feizi. Identifying interpretable subspaces in image representations. In International Conference on Machine Learning, pp. 15623–15638. PMLR, 2023.

[2] Tue M Cao, Nhat X Hoang, Hieu H Pham, Phi Le Nguyen, and My T Thai. Neurflow: Interpreting neural networks through neuron groups and functional interactions. ICLR, 2025.

---

> ### Author Response · Authors · 2025-12-03
>
> Thank you very much for your thoughtful and positive review of our work, and for highlighting the strengths of our activation-based basket construction, the consistency metric, and the idea of set-operation–based reasoning over neuron groups. Below, we have revised the manuscript and appendix accordingly.
>
> ---
>
> W2. interest in separability between neuron groups
>
> We thank the reviewer for pointing out the importance of inter-group criteria such as *separability*. We agree that consistency captures only the intra-group relationship, while separability characterizes how distinct different groups are from each other.
>
> We would like to clarify that our cluster selection already incorporates a simplified Davies–Bouldin Index (DBI), which explicitly depends on both within-cluster dispersion and between-cluster separation. In other words, the DBI we use in our filtering step is designed to retain clusters that are internally compact and well separated from other clusters, thereby addressing both consistency and separability at the cluster level.
>
> In the revised Appendix B.1, we report the average DBI scores of the selected clusters, along with comparisons to clusters that are rejected by our filtering. These results clarify that the proposed selection procedure already optimizes a criterion that combines intra-group consistency with inter-group separability, in line with your suggestion.
>
> ---
>
> W3. Additional support for filtering metrics and their effect
>
> You asked how the proposed filtering metrics affect the quality of the neuron groups and how aggressive the filtering is.
>
> - **How many clusters are removed?**
>   At ResNet-50 layer4, level-1 we obtain 207,902 valid clusters. Of these, 199,716 clusters (96.06%) pass *all three* criteria (cohesion, purity, DBI). The per-metric failure rates are very small (cohesion: 0.26%, purity: 0.61%, DBI: 3.20%). Thus, we intend to keep clusters; filtering removes only a small tail of outliers.
>
> - **What is removed?**
>   In the revised Appendix E, we show representative filtered clusters for each criterion:
>   - *Low cohesion / purity*: visually noisy or mixed clusters, e.g., “blue bird eye” patches mixed with red berries and toys, or lobster patches mixed with human legs and clothes. These have no single coherent meaning even for humans.
>   - *High DBI*: near-duplicate clusters (e.g., two separate groups of nearly identical “finger” patches or “black fur” patches) that are not meaningfully separable; dropping one does not reduce concept coverage, but avoids redundancy.
> ---
>
> W4 Usefulness of set-operation–based reasoning
>
> We appreciate your positive remarks on our proposal of “set-theoretic reasoning.” To avoid over-claiming, we have revised the wording throughout the paper to describe our analysis as set-operation–based (union/difference) rather than fully formal set theory over neuron indices.
>
> To further support this analysis, we added a parent–child study (App. D) using randomly sampled parent–child pairs from the hierarchy:
>
> - For level 1–2 in ResNet-50, when we explicitly take the union of child baskets and compare it to the corresponding parent basket, we find:
>   - mean parent coverage ≈ 0.94,
>   - cosine similarity between parent and child-union activation profiles ≈ 0.995 ± 0.006.
>
> These results indicate that parent baskets are very well approximated by the union of their children in activation space, providing quantitative backing that our union/difference operations reflect a meaningful compositional structure, beyond the qualitative examples in the paper.
>
> ---
>
> W5 Effect of the number of clusters on the consistency metric (minor)
>
> Thank you for raising this point. To check whether the different number of groups biases the metric, we re-ran our method by restricting to 344 randomly sampled baskets, and repeated this 5 times.
>
> - With 10,000 baskets (original setting): consistency = 0.7404 ± 0.0004.
> - With 344 randomly sampled baskets (5 runs): consistency = 0.7417 ± 0.0039.
>
> The scores are nearly identical, suggesting that the consistency measure is essentially insensitive to the number of baskets used in our method, and that our comparison with [1] is not driven by the difference in cluster count. We clarify this experimental detail in the revised appendix.

---

### Author Response · Authors · 2025-12-03

Dear Area Chair,

We thank you and the reviewers (BvRj, 7sy9, B6Vw, fGnd) for the constructive feedback.
We have revised the manuscript and appendix accordingly.

In brief, Neurobasket introduces a basket-first framework that groups neurons into hierarchical, activation-consistent baskets by clustering image activations and selecting stable responders. On top of this, we performed set-operation–based analysis (union/difference) to link combinations of neuron groups that contribute to model predictions. We quantitatively and qualitatively validate that baskets are stable, semantically aligned with their images, and are also core contributors for each concept.

---
### 1. Strengths Highlighted by Reviewers

| Category | Highlighted strength (as noted by reviewers) | Reviewers |
|---------|-----------------------------------------------|-----------|
| Problem & motivation | Points out a real limitation of single-neuron explanations when representations are distributed, and tackles an important, timely problem in interpretability. | B6Vw, fGnd |
| Basket construction & metrics | Provides concrete, activation-based criteria for neuron selection in each basket and introduces a consistency metric for comparing group quality. | BvRj |
| Hierarchy & set-operation view | Builds a hierarchy over neuron groups and uses union/difference style operations as a useful conceptual tool for analyzing representations. | BvRj, B6Vw, fGnd |
| Experiments & reproducibility | Evaluates across multiple backbones/datasets with reasonable experimental design, random baselines, detailed appendix, and released code supporting reproducibility. | fGnd, 7sy9 |
---


### 2. Resolution of Reviewer Concerns
We summarize how our revisions specifically address the key concerns raised by the reviewers.

| Key Concern | Resolution (Rebuttal Actions) | Reviewers |
|-------------|--------------------------------|-----------|
| Quantitative evaluation of semantic faithfulness / interpretability | Added two embedding-based checks: (i) Sentence-BERT sim between basket summaries and their own image captions vs. random, (ii) CLIP image–text sim between basket images and summaries vs. random; both show clear improvements over random. | fGnd, 7sy9 |
| Validity of “set-theoretic reasoning” and parent–child relation | Clarified that we perform **set-operation–based** analysis (union/difference), not strict set algebra. Added parent vs. child-union study showing high parent coverage (\~0.94) and activation-profile cosine (\~0.995), supporting compositionality in practice (App. D). | fGnd, 7sy9, BvRj |
| Relation to concept directions / SAEs and novelty of Neurobasket | Expanded Related Work to position CAV/CRAFT and SAEs/Matryoshka SAEs as direction/latent–space methods, while Neurobasket builds a basket-first, neuron-index–level hierarchy with set-operation–based causal analysis. Clarified complementarity and distinction from SNE (CLIP-Dissect, WWW) and MNE (HINT, NeurFlow). | fGnd, B6Vw |
| Sensitivity of neuron-selection hyperparameters α and β | Added α/β ablations (App. B.3) showing smooth trade-off: higher thresholds lead to fewer, more consistent baskets but smaller mean Δlogit\_gt; our settings (α=3, β=0.75) balances internal consistency and causal impact, all settings remain well above random baselines. | 7sy9, fGnd, B6Vw |
| Effect of filtering on completeness of explanations | Reported that filtering is very conservative (e.g., ResNet-50 layer4 level-1 keeps 96.1% of 207,902 clusters; each metric fails <4%). Added examples showing removed clusters are noisy/mixed or near-duplicate, indicating we prune outliers rather than core concepts (App. E). | fGnd, B6Vw |
| Abstraction claims for Fig. 6 | Added GPT-based abstraction scores for images and captions across depth, showing consistent increases from shallow to deep blocks for ResNet-50/18 and ViT, quantitatively supporting the “more abstract at deeper layers” claim (App. F). | 7sy9, fGnd |
| Comparison with other SNE/MNE baselines |  We clarified our position that Neurobasket is different from existing SNE/MNE methods, which is a hierarchical, and set-operation–based framework for understanding distributed neural representations. | 7sy9, B6Vw, fGnd |
---

We appreciate your time and consideration. We hope that, with these revisions, the paper now offers
a clearer and more thoroughly supported view of Neurobasket as a stable, hierarchical, and
set-operation–based framework for understanding distributed neural representations.

---

### Meta-Review · Area_Chair_koHU · 2026-01-04

**Summary:**

Across reviewers, the main concerns center on limited novelty, insufficient quantitative evaluation, and unclear validation of interpretability claims.

a) The idea of hierarchically grouping neurons and grounding these groups with textual descriptions is viewed as conceptually appealing but largely incremental relative to prior work on hierarchical neuron grouping and multi-neuron explanations (e.g., HINT, NeurFlow, FALCON), as well as related approaches based on feature directions (e.g., CRAFT, sparse autoencoders).

b) Reviewers consistently note that the evaluation relies heavily on qualitative examples and a single notion of “consistency”, without adequately assessing other important properties such as separability, faithfulness, compositionality, or causal relevance, nor comparing against a broader set of established baselines. The proposed claims of set-theoretic reasoning (e.g., union, difference, hierarchy) are considered under-validated, with ambiguity as to whether these operations are truly compositional or merely illustrative, lacking quantitative evaluations in conventional XAI settings or downstream tasks.

c) In addition, the interpretability pipeline depends critically on VLM/LLM-generated captions and summaries, raising concerns about reproducibility, stability, and semantic faithfulness due to model non-determinism and hallucination.

d) Reviewers highlight insufficient analysis of key hyperparameters and filtering steps, which may significantly affect the completeness and robustness of the resulting neuron group explanations.

**Reviewer Concerns:**

The authors addressed several reviewer concerns in the rebuttal. In particular, they clarified that the proposed basket construction operates directly in activation and probability space and does not rely on VLM- or LLM-generated captions or summaries, thereby mitigating concerns regarding semantic faithfulness, non-determinism, and dependence on language models. In addition, the authors provided further quantitative analyses examining semantic faithfulness and interpretability, sensitivity to the neuron-selection hyperparameters $\alpha$ and $\beta$, and the impact of the filtering step. These additions partially address reviewer concerns regarding robustness and sensitivity (corresponding to points (c) and (d) in the previous section).

However, key concerns remain insufficiently resolved. In particular, issues related to technical novelty and the lack of clear quantitative advantages over existing SNE and MNE baselines persist. Although the authors argue that their framework does not rely on predefined concepts and enables set-operation-based analysis of neuron groups, the hierarchical construction of baskets itself is not fundamentally new relative to prior work. Moreover, I concur with Reviewer fGnd that the evaluations presented in both the paper and the rebuttal — largely centered on a single metric (activation consistency) — are not sufficient to substantiate claims of improved interpretability (or downstream utility). More comprehensive evaluations, including those aligned with established XAI benchmarks and settings, would be necessary to convincingly demonstrate the benefits of the proposed approach.

**Reviewer Scores:**

(a) Reviewer BvRj would likely maintain their original score, with a possibility of increasing it to 6, as their primary concerns regarding model sensitivity to hyperparameters and the filtering strategy were addressed in the rebuttal.
(b) Reviewer 7sy9 would likely maintain their original score, as they did not appear to be fully engaged in/beyond their initial review.
(c) Reviewer B6Vw would likely maintain their original score, since their main concerns pertain to technical novelty, which was only partially addressed in the rebuttal.
(d) Reviewer fGnd actively participated in the discussion and may increase their score to 4, though their overall assessment would likely remain on the negative side.

---

### Decision · Program_Chairs · 2026-01-26

Reject